# Synergistic recruitment of UbcH7~Ub and phosphorylated Ubl domain triggers parkin activation

Tara EC Condos[1,†], Karen M Dunkerley[1,†], E Aisha Freeman[1,†], Kathryn R Barber[1], Jacob D Aguirre[1], Viduth K Chaugule[2], Yiming Xiao[3], Lars Konermann[3], Helen Walden[2] & Gary S Shaw[1,*]

## Abstract

The E3 ligase parkin ubiquitinates outer mitochondrial membrane proteins during oxidative stress and is linked to early-onset Parkinson's disease. Parkin is autoinhibited but is activated by the kinase PINK1 that phosphorylates ubiquitin leading to parkin recruitment, and stimulates phosphorylation of parkin's N-terminal ubiquitin-like (pUbl) domain. How these events alter the structure of parkin to allow recruitment of an E2~Ub conjugate and enhanced ubiquitination is an unresolved question. We present a model of an E2~Ub conjugate bound to the phospho-ubiquitin-loaded C-terminus of parkin, derived from NMR chemical shift perturbation experiments. We show the UbcH7~Ub conjugate binds in the open state whereby conjugated ubiquitin binds to the RING1/IBR interface. Further, NMR and mass spectrometry experiments indicate the RING0/RING2 interface is re-modelled, remote from the E2 binding site, and this alters the reactivity of the RING2(Rcat) catalytic cysteine, needed for ubiquitin transfer. Our experiments provide evidence that parkin phosphorylation and E2~Ub recruitment act synergistically to enhance a weak interaction of the pUbl domain with the RING0 domain and rearrange the location of the RING2(Rcat) domain to drive parkin activity.

**Keywords** dynamics; E2 conjugating enzyme; E3 ubiquitin ligase; Parkinson's disease; ubiquitination
**Subject Categories** Post-translational Modifications, Proteolysis & Proteomics; Structural Biology
**The EMBO Journal (2018) 37: e100014**

## Introduction

Parkinson's disease is the second most common neurodegenerative disease estimated to affect 1% of the population over 60 years of age (Tysnes & Storstein, 2017). The disease is believed to be a result of genetic predisposition or environmental factors (Corrigan et al, 1998) that lead to oxidative damage of mitochondrial proteins (Alam et al, 1997) and subsequent mitochondrial dysfunction (Schapira et al, 1990) and is characterized in patients by the loss of dopaminergic neurons in the substantia nigra of the midbrain (Hornykiewicz, 1966; Riederer & Wuketich, 1976). In addition to sporadic Parkinson's disease, there are also genetic forms of the disease that account for approximately 10% of all cases. In particular, mutations in the genes for *PARK2* and *PARK6* give rise to early-onset or autosomal recessive juvenile parkinsonism (ARJP) forms of the disease that have similar symptoms including rigidity, bradykinesia and postural instability (Jankovic, 2008) but affect individuals at a much younger age. *PARK2* encodes the E3 ubiquitin ligase parkin (Kitada et al, 1998) where mutations account for 50% of all ARJP cases. Along with the PTEN-induced kinase (PINK1) translated from *PARK6*, these proteins use the ubiquitin degradation pathway to turnover damaged mitochondria and maintain mitochondrial homeostasis, especially under conditions of oxidative stress.

Parkin is a member of the RBR E3 ligase family that also includes the human homolog of Ariadne (HHARI) and HOIL-1 interacting protein (HOIP; Spratt et al, 2014). These enzymes have a characteristic RBR motif comprising *R*ING1, in-*B*etween-RING and *R*ING2 (Rcat) domains that distinguish them from HECT and RING classes of E3 enzymes in terms of structure, mechanism and functionality. In particular, RBR E3 ligases incorporate a hybrid ubiquitination mechanism (Wenzel et al, 2011) whereby an E2 conjugating enzyme is recruited to the RING1 domain (similar to RING E3 ligases) and ubiquitin (Ub) is transferred from the E2~Ub conjugate to a catalytic cysteine in the RING2(Rcat) domain (similar to HECT E3 mechanisms) prior to labelling of a substrate lysine. RBR E3 ligases and RING E3 ligases have RING domains that are structurally similar and are expected to recruit E2 enzymes in a similar fashion (Budhidarmo et al, 2012), as recently shown in crystal structures of the RBR E3 ligases HHARI with UbcH7-Ub (Dove et al, 2017; Yuan et al, 2017) and HOIP with UbcH5b-Ub (Lechtenberg et al, 2016). However, a distinguishing feature of the HHARI and HOIP RBR E3 ligases is their ability to recognize an extended ("open") form of the E2~Ub conjugate similar to that used by HECT E3 enzymes. This E2~Ub arrangement promotes a conformation susceptible to the

1   Department of Biochemistry, The University of Western Ontario, London, ON, Canada
2   Institute of Molecular Cell and Systems Biology, University of Glasgow, Glasgow, UK
3   Department of Chemistry, The University of Western Ontario, London, ON, Canada
    *Corresponding author. Tel: +1 519-661-4021; Fax: +1 519-661-3175; E-mail: gshaw1@uwo.ca
    †These authors contributed equally to this work

transthiolation reaction needed to transfer the Ub cargo from the E2 enzyme to the catalytic cysteine of the RING2(Rcat) domain. Parkin on the other hand has been shown to function with a variety of E2 enzymes including UbcH7 and UbcH5b (Chaugule *et al*, 2011; Wenzel *et al*, 2011) although it appears that UbcH7 is the optimal E2 enzyme owing to its preference for ubiquitin transfer to cysteine, a requirement for RBR E3 ligases. While the conformation of the UbcH7~Ub conjugate during recruitment by parkin is unknown, it has been established that a cryptic Ub binding site within the RING1–IBR interface is only uncovered upon pUb binding to parkin and this has been proposed to help coordinate E2~Ub recruitment (Kumar *et al*, 2017).

All RBR E3 ligases identified to date, including parkin, appear to be uniquely regulated (Spratt *et al*, 2014; Walden & Rittinger, 2018). Parkin is normally autoinhibited by an accessory ubiquitin-like (Ubl) domain (Chaugule *et al*, 2011) that blocks both the E2 and cryptic ubiquitin sites. In addition, structures show that another accessory module, the RING0 domain, partially obscures the catalytic cysteine in the RING2(Rcat) domain protecting this site from Ub transfer. At least two steps have been identified for the activation of parkin both as a result of phosphorylation by PINK1. Under oxidative stress conditions, PINK1 is activated and phosphorylates ubiquitin (pUb) near the outer mitochondrial membrane. This in turn helps recruit parkin to the membrane through binding of pUb to the RING1–IBR region of the E3 ligase (Sauvé *et al*, 2015; Wauer *et al*, 2015; Kumar *et al*, 2017) and subsequent phosphorylation of parkin's Ubl (pUbl) domain (Ordureau *et al*, 2014). These two events greatly stimulate ubiquitination activity (Kondapalli *et al*, 2012; Shiba-Fukushima *et al*, 2012; Kane *et al*, 2014; Kazlauskaite *et al*, 2014; Koyano *et al*, 2014) through an allosteric displacement mechanism of the pUbl domain from parkin (Kumar *et al*, 2015; Sauvé *et al*, 2015). What is less clear is how parkin positions the E2~Ub conjugate to enable transfer of the Ub molecule to the RING2 (Rcat) domain as a necessary step for catalysis. Current crystal structures of parkin show the proposed E2 binding site on the RING1 domain is > 50 Å from the catalytic site (C431) in the RING2 (Rcat) domain suggesting a significant conformational rearrangement is needed (Riley *et al*, 2013; Trempe *et al*, 2013; Wauer & Komander, 2013; Kumar *et al*, 2015, 2017; Sauvé *et al*, 2015; Wauer *et al*, 2015). A similar dilemma arises from recent structures of HHARI in complex with UbcH7-Ub that show the ubiquitin molecule is 47–53 Å from the catalytic site (Dove *et al*, 2017; Yuan *et al*, 2017). Alternatively, structures of HOIP:E2-Ub and parkin:pUb complexes assembled from domain-swapping or symmetry-related molecules raise the possibility of co-operation between multiple E3 ligase molecules to promote ubiquitin transfer (Lechtenberg *et al*, 2016; Kumar *et al*, 2017).

In this work, we identify how the pUbl domain and E2~Ub conjugate co-operate to regulate parkin activity. We use NMR spectroscopy and chemical shift perturbation experiments to determine a model of pUb-bound parkin in complex with its biological UbcH7~Ub conjugate. The structure shows that a non-hydrolysable UbcH7-Ub conjugate binds in an altered "open" conformation with its thioester linkage poised for Ub transfer to the catalytic cysteine of the RING2(Rcat) domain. We show that E2-Ub recruitment to parkin results in two distinct types of NMR chemical shift changes: one set that is consistent with the E2~Ub binding site and a second set that corresponds to residues near the RING0/RING2 interface,

indicating this region is re-modelled during E2~Ub recruitment. We show that the reactivity of the catalytic cysteine (C431) in the RING2(Rcat) domain is sensitive to both parkin phosphorylation and E2-Ub binding. Further, we use NMR spectroscopy and hydrogen–deuterium exchange (HDX) mass spectrometry experiments to show that the pUbl domain undergoes transient interaction with the RING0 domain that is enhanced upon E2~Ub recruitment by parkin. These events also result in large changes in the exposure of the RING2(Rcat) domain consistent with its rearrangement. Overall, our work provides a dynamic picture of parkin activation whereby PINK1 phosphorylation of parkin and E2~Ub recruitment co-operate to drive parkin activity.

# Results

## Dual phosphorylation leads to dynamic repositioning of the pUbl domain

Multiple high-resolution structures of parkin have provided significant insights into its E3 ligase function. Structures of near full-length parkin (Kumar *et al*, 2015; Sauvé *et al*, 2015) show the C-terminal RING0–RING1–IBR–RING2(Rcat) domains (termed "R0RBR") form a compact unit whereby the N-terminal Ubl domain interacts with the RING1 and IBR domains and portions of the tether region (Fig 1A). This mode of interaction confirmed that the Ubl domain exerted its previously identified autoinhibitory effect (Chaugule *et al*, 2011) by blocking the expected binding site on the RING1 domain from an E2 conjugating enzyme resulting in negligible ubiquitination activity. Despite the apparent well-folded, compact nature of parkin in this state, a wealth of flexibility exists within the E3 ligase that is not obvious from the crystal structures. NMR dynamics experiments show significant mobility for the IBR domain and segments on either side of a short helix within the tether region where poor electron density is frequently observed in crystal data (see fig S4 in Kumar *et al*, 2015). Details of the partial E3 ligase activation of parkin have been shown in complexes with phosphorylated ubiquitin (pUb; Kumar *et al*, 2015, 2017; Wauer *et al*, 2015), where pUb binds to a broad crevasse between the RING0 and RING1 domains (Fig 1A). The association of pUb causes rearrangement of the IBR domain and results in the formation of a large gap between the IBR and RING1 domains. This results in decreased mobility of the IBR domain due to its juxtaposition with the pUb molecule as shown by heteronuclear nOe experiments (Fig EV1) and increased mobility in the tether region, especially residues V380-A390 that immediately follow the IBR domain. In this optimized state, PINK1 is able to phosphorylate the Ubl domain more efficiently (Ordureau *et al*, 2014; Aguirre *et al*, 2018) resulting in nearly 10-fold decreased affinity of the pUbl domain and increased affinity for pUb at the RING0/ RING1 interface. This allosterically releases the pUbl domain from its interaction with the RING1 domain (Kumar *et al*, 2015; Sauvé *et al*, 2015; Aguirre *et al*, 2017) although no structures of this complex for full-length parkin exist. To assess this event, the structure and dynamics of autoinhibited parkin and those of phosphorylated parkin in complex with pUb (pParkin:pUb) were probed by HDX experiments measured by mass spectrometry (Fig 1B–D and Appendix Fig S1). These experiments were used to provide a measure of the accessibility of the backbone amides to solvent and

the strength of hydrogen bonding in parkin. Based on previous structural data, these experiments should probe at least three aspects of parkin activation: (i) the binding site for the pUb molecule, (ii) changes in the position of the Ubl domain upon phosphorylation and (iii) indirect structural changes that result from either of these events. The HDX data, which show 91% coverage for parkin (Appendix Fig S2), highlight that many regions of the RING0 and RING1 domains exhibit the largest differences in their HDX properties (Fig 1C). For example, a series of peptides covering regions of the RING0 (Y147-Q155, F209-A225) and RING1 (F277-L283, I298-E309, Y312-E322, V324-L331) domains show slower exchange in the pParkin:pUb state due to protection of these regions by pUb binding at the RING0/RING1 interface (Fig 1D). Coincident with this, peptides at the extreme C-terminus of the IBR domain (C365-A379), the tether region (Y391-Q400) and nearly the entire Ubl domain show increased deuterium exchange (Fig 1C and D). One region of the Ubl domain (R42-E49) shows slower exchange upon phosphorylation. This is likely a direct effect of phosphorylation only since several residues in this region (I44, A46, G47) have decreased amide exchange in the isolated Ubl domain (Aguirre et al, 2017). These observations show the pUbl domain is no longer bound at the IBR/RING1 interface and, consistent with previous NMR, sedimentation velocity and computational experiments, indicates the pUbl domain adopts a range of bound/free conformations in the pParkin:pUb state (Fig 1D; Caulfield et al, 2014; Aguirre et al, 2017). In this scenario, release of the pUbl domain exposes the predicted E2~Ub binding site, based on other RING/E2 complexes (Lechtenberg et al, 2016; Dove et al, 2017; Yuan et al, 2017).

## The Parkin/UbcH7-Ub complex reveals an E2-Ub conformation poised for Ub transfer

Phosphorylation of Ub, its recruitment to parkin and subsequent phosphorylation of the Ubl domain result in an increased affinity for the UbcH7~Ub conjugate (Kumar et al, 2015) used by parkin to efficiently transfer ubiquitin to the RING2(Rcat) domain and propagate the ubiquitination reaction. As a first step to identify how UbcH7~Ub stimulates the transfer of its Ub cargo to the catalytic cysteine within the RING2(Rcat) domain, we used NMR spectroscopy to examine how the E2~Ub complex is recruited to the R0RBR C-terminal region of parkin. This work required multiple modes of isotopic labelling, chemical shift assignment of the selected proteins and in some cases purification of the assembled complexes. For example, the chemical shift assignment of R0RBR parkin was completed using TROSY-based triple-resonance experiments using $^2$H,$^{13}$C,$^{15}$N R0RBR parkin (Kumar et al, 2015). A non-hydrolysable E2~Ub conjugate was assembled using a triple-substituted UbcH7 protein (UbcH7$^{C17S/C86K/C137S}$) to allow formation of the isopeptide linkage between the UbcH7 and Ub (UbcH7-Ub) and prevent oxidation of the E2 enzyme during NMR experiments. Due to the size of the complexes formed, which compromises the quality of NMR spectra, we utilized mixtures containing a triple-labelled ($^2$H,$^{13}$C,$^{15}$N) protein titrated with a deuterated partner. For example, $^2$H,$^{13}$C,$^{15}$N-labelled R0RBR parkin was complexed with invisible $^2$H-labelled pUb and purified to homogeneity by size-exclusion chromatography (R0RBR:pUb) prior to the addition of $^2$H-labelled UbcH7-Ub conjugate.

Initial chemical shift perturbation experiments monitored the $^1$H–$^{15}$N TROSY spectra of the $^2$H,$^{13}$C,$^{15}$N-labelled R0RBR:$^2$H-labelled pUb complex with $^2$H-labelled UbcH7-Ub. These data show that many resonances in R0RBR parkin shift in the presence of E2-Ub and undergo line broadening indicative of a nearly 70-kDa complex being formed (Fig 2A). Formation of the UbcH7-Ub complex with R0RBR:pUb occurs with slow exchange binding kinetics indicative of a $K_d < 1$ μM measured previously by isothermal titration calorimetry (Kumar et al, 2015, 2017). Surprisingly, spectral changes were obvious in two distinct regions of R0RBR—one region where residues are mostly surface exposed and a second region where residues are largely buried (Figs 2B and EV2). The first area consists of resonances belonging to surface residues within the RING1, IBR and tether region indicative of a canonical E2/RING E3 binding interface as well as the Ub interaction site. Resonances affected by E2 binding were located in L1 (C241, T242), helix H1 (L266, V269-L272), L2 (C293) and the tether (Q389-R392, D394-R396) regions of R0RBR while residues in the IBR domain (V330-R334) and tether (A379-F381) corresponded to the Ub binding region (Figs 2B and EV2A). The E2 binding region was verified through separate NMR experiments that monitored the individual binding of $^2$H-labelled UbcH7 with $^2$H,$^{13}$C,$^{15}$N-labelled R0RBR (Fig EV2B). This experiment showed more limited chemical shift changes but included distinctive line broadening and loss of signals for the L1, helix H1 and L2 regions of the RING1 domain confirming the E2 interaction site with little effect on resonances from residues in the IBR domain.

Reciprocal NMR experiments were conducted to identify the binding surfaces for the UbcH7 and Ub components within the UbcH7-Ub conjugate upon binding to the R0RBR:pUb complex. In these experiments, we titrated size-exclusion purified $^2$H-R0RBR:$^2$H-pUb complex into a solution of $^2$H,$^{15}$N-labelled UbcH7-Ub conjugate and monitored chemical shift changes using $^1$H–$^{15}$N TROSY NMR spectroscopy (Appendix Fig S3). These experiments were complicated because UbcH7-Ub predominantly forms a closed state in the absence of a binding partner, but reverts to an open state upon binding to an RBR E3 ligase (Dove et al, 2016). Thus, we expected to see chemical shift changes in UbcH7-Ub that reflected both conversion to the open state and binding to R0RBR:pUb. The closed state of UbcH7-Ub was easy to identify based on previous experiments (Dove et al, 2016). Initial $^1$H–$^{15}$N HSQC spectra of the UbcH7-Ub conjugate showed numerous resonances in UbcH7 (F22, V40, N43, N56, K100-N113) and Ub (G47-L50) that reflect close proximity between helix H2 in UbcH7 and Ub (Appendix Fig S4). Upon binding to R0RBR:pUb, most of these signals return to a similar position as found in the unconjugated (free) UbcH7 protein indicating Ub is not occupying the closed position. These observations indicate that parkin converts the UbcH7-Ub conjugate from the predominantly closed state to a more open conformation, a similar observation as made for the RBR E3 ligase HHARI (Dove et al, 2016). Indirectly this also shows the helix H2 region (K100-N113) in UbcH7 and the K48 loop in Ub are not at the R0RBR interface. Thus these chemical shift changes were not considered to map the binding interface in the complex. In contrast to the changes observed for conversion of the closed to the open states of UbcH7-Ub, titration of the R0RBR:pUb complex into the UbcH7-Ub conjugate resulted in broadening of many resonances. Notably, several signals in both UbcH7 and

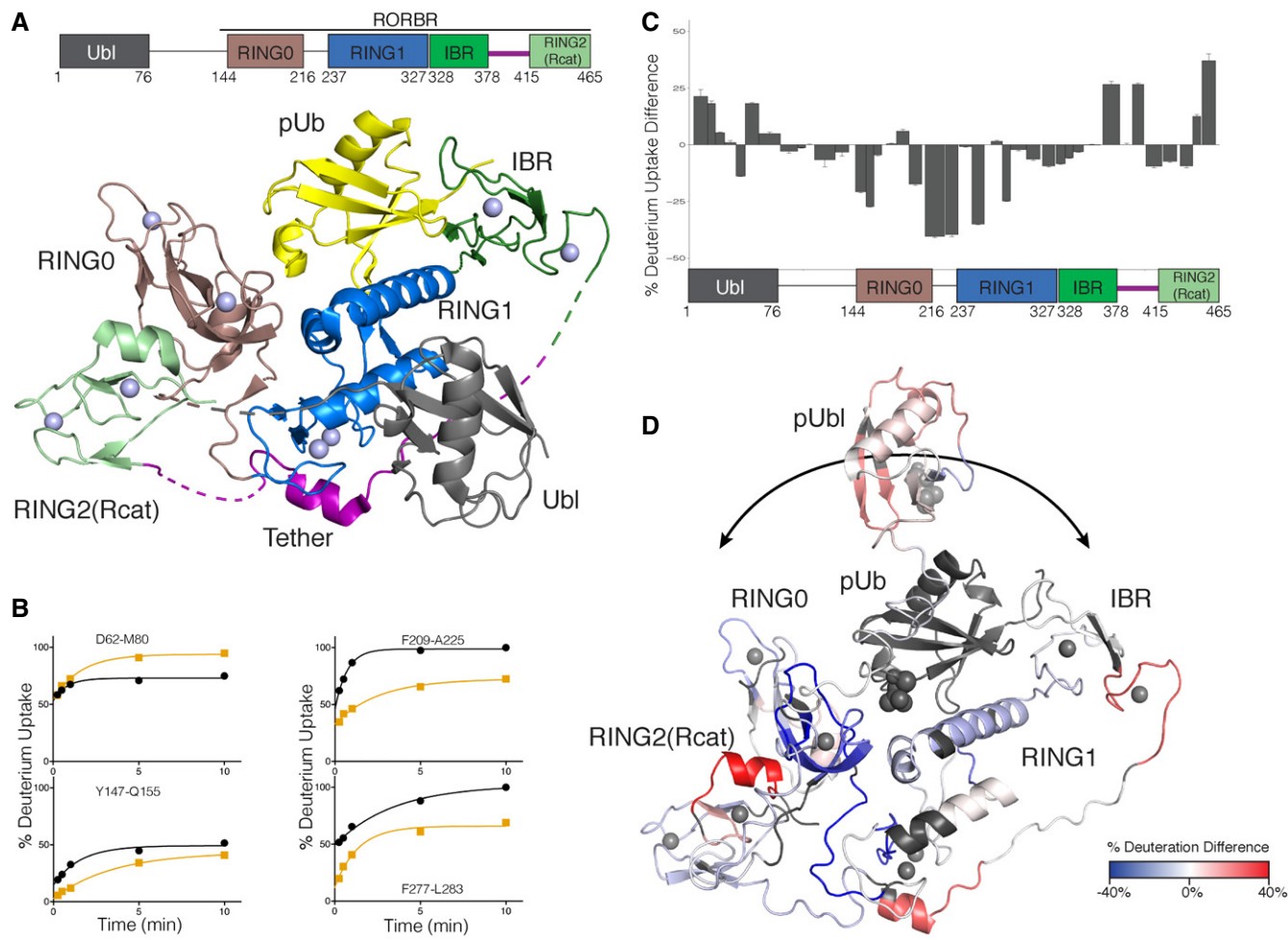

**Figure 1. Autoinhibited and released states of pUbl prior to E2~Ub recruitment.**

A  Cartoon structure of optimized, inhibited parkin from X-ray crystal structure (PDB 5N2W; Kumar *et al*, 2017). The structure shows the Ubl (grey), RING0 (brown), RING1 (blue), IBR (green) and RING2(Rcat; pale green) domains along with the tether region (purple). A schematic block diagram is shown above to illustrate the domains. The pUb (yellow) molecule is shown bound to the RING0/RING1 interface. Zn atoms are shown as spheres, and regions of parkin not observed in the crystal structure are indicated as dashed lines. The cartoon figure was created using PyMol (Delano, 2002).

B  Selected examples of % deuterium uptake measured by HDX mass spectrometry to show differences between full-length autoinhibited parkin (black lines) and phosphorylated parkin (pParkin) in complex with pUb (orange lines). Each plot shows the % deuterium uptake for the peptide indicated as a function of time.

C  Differences in % deuterium uptake between autoinhibited parkin and pParkin:pUb state are plotted at the one-minute time point. The width of the bars shown represents the length of the peptides observed and measured, while the heights represent the relative difference in hydrogen–deuterium exchange between the two species. Error bars represent standard deviation above the average for triplicate measurements. Positive bars indicate a greater deuterium uptake in the pParkin:pUb state compared to the autoinhibited state indicating greater exposure to solvent and/or weaker hydrogen bonding, while a negative change indicates the pParkin site is more protected. The domain structure of parkin is shown below the data to indicate the location of peptides.

D  Relative HDX differences are mapped to the structure of parkin:pUb (PDB 5N2W) where the pUbl domain has been arbitrarily positioned away from the remainder of the protein. Regions where HDX was faster in the pParkin:pUb complex relative to parkin alone at the one-minute time point are indicated in red, while those regions in blue exchange more slowly using the gradient shown. Several regions not visible in crystal structures were modelled into the structure using the Modeller tool (Eswar *et al*, 2006) in Chimera (Pettersen *et al*, 2004). The arrows at the top of the figure indicate the pUbl domain samples multiple conformations based on its overall faster HDX in the phosphorylated state and previous NMR and analytical sedimentation velocity experiments (Aguirre *et al*, 2017).

Ub underwent significant chemical shift changes or shifted such that they could not be identified in spectra, where the UbcH7-Ub conjugate was saturated with the R0RBR:pUb complex, due to slow or slow–intermediate chemical shift exchange. These resonances correspond to those in helix 1 (R6, M8 and L11), loop 4 (A59, F63) and loop 7 (N94, K96, A98) of UbcH7 and the β1-β2 loop (V5, T7, L8, T9 and I13), the linker following helix α1 (E34, I36) and C-terminus (V70, G75) of Ub (Appendix Fig S3).

We used the results from chemical shift perturbation experiments to determine a model of the UbcH7-Ub conjugate bound to R0RBR:pUb using HADDOCK (Dominguez *et al*, 2003). This was done using the crystal structures of R0RBR:pUb (PDB 5N2W), UbcH7 (PDB 4Q5E) and Ub (PDB 1UBQ) as starting points and imposing distance restraints between the proteins to conduct three-molecule docking of UbcH7 and Ub to the R0RBR:pUb complex. Distance restraints were selected for residues that became unobservable or shifted more

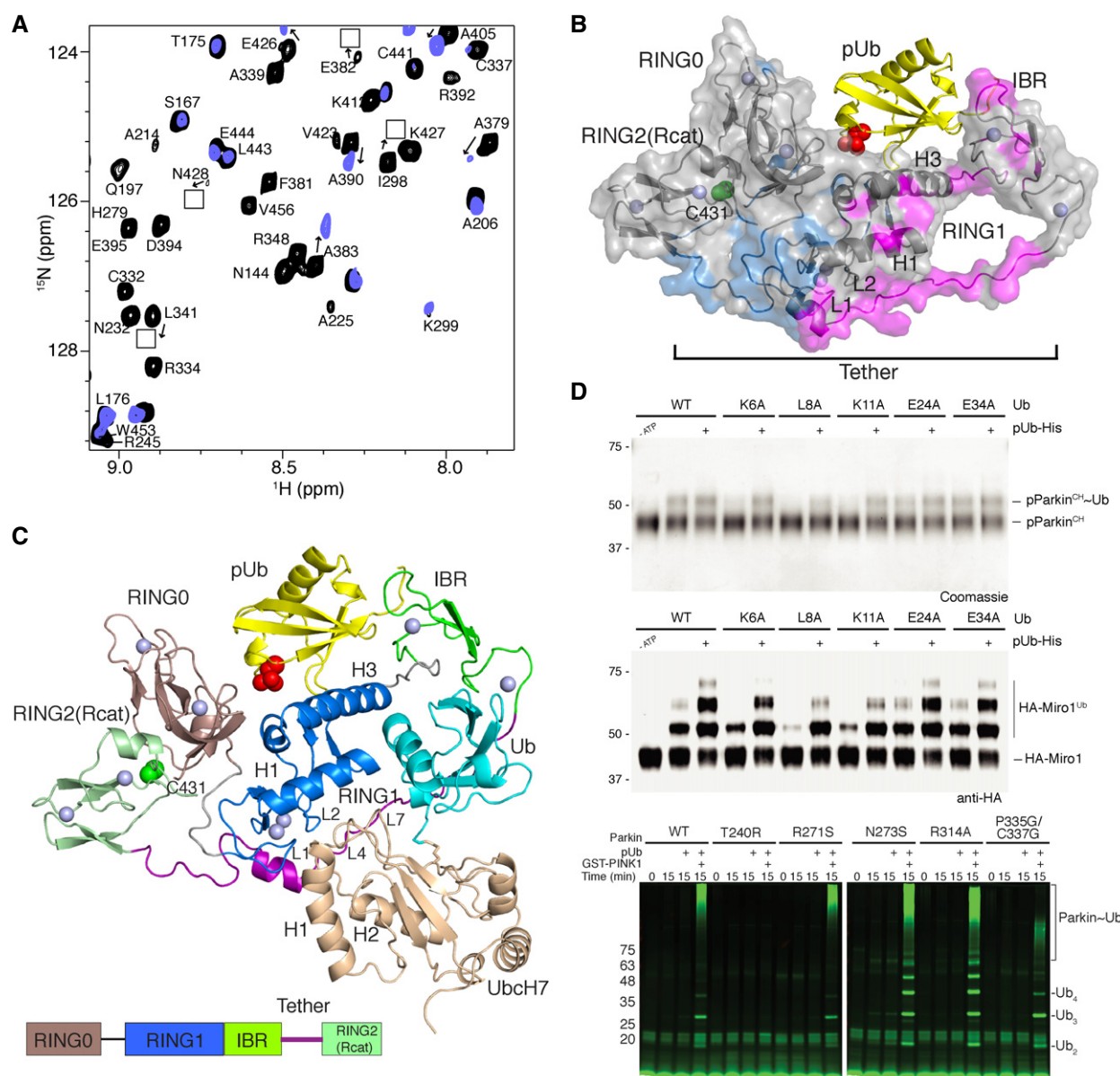

Figure 2.

than one standard deviation above the average shift (Fig EV2 and Appendix Fig S3), and had a side chain surface exposure > 20%. We also included neighbouring residues to those with large chemical shift changes that were solvent exposed ("passive" residues according to the HADDOCK protocol). This resulted in a total of 23 ambiguous distance restraints between R0RBR and the UbcH7 moiety and 25 ambiguous restraints between the R0RBR and Ub proteins. A single unambiguous restraint was used to mimic the isopeptide bond between the carboxylate in G76 of Ub and the side chain amine from K86 in UbcH7 during calculations. The resulting models showed the location of the UbcH7 and Ub proteins with respect to R0RBR:pUb is similar in all 100 water-refined complexes. The best 20 complexes have a backbone RMSD of 0.71 ± 0.10 for R0RBR:pUb:UbcH7-Ub. Comparison of the structures shows that some variation in the orientation of the Ub molecule is observed

that was not evident for R0RBR, pUb or UbcH7 proteins in the models. This may indicate the Ub protein in the UbcH7-Ub complex is more dynamic (Fig EV3) than the remainder of the complex. In the lowest energy structure (Fig 2C), the UbcH7-Ub conjugate takes on an open conformation in the complex where the UbcH7 moiety interacts mostly with the RING1 domain and the tether region while the Ub molecule interacts with the RING1/IBR pocket and the N-terminus of the tether region. In the complex, helix H1 (R6, K9) and loop L4 (E60, F63-K64) in UbcH7 contact the R0RBR loop L1 (T240, T242) and helix H1 (L266, T270, R271, D274) respectively in RING1 (Fig 2C). As with other E2/RBR E3 ligase complexes (Lechtenberg *et al*, 2016; Dove *et al*, 2017; Yuan *et al*, 2017), the L7 loop (N94, K96, P97, A98) in UbcH7 sits adjacent to loop L2 (V290-G292) in RING1, but has additional contacts with Y391-D394 just prior to the short helix in the tether region. We noted that some of the largest

◀

**Figure 2.  Model of the E2-Ub conjugate bound to pUb-activated parkin.**

A  Portion of the $^1H$–$^{15}N$ TROSY NMR spectrum showing signals from R0RBR parkin within the R0RBR:pUb complex (black) and in the presence of one equivalent of the unlabelled UbcH7-Ub conjugate (blue). Signals that shift in the presence of UbcH7-Ub are indicated by arrows. Some signals shift and broaden and cannot be identified in the bound state. Boxes indicate signals that have shifted and are visible at very low contour level.

B  Two distinct surfaces are revealed on parkin upon binding of the UbcH7-Ub conjugate. Chemical shift perturbations were measured from $^1H$–$^{15}N$ TROSY spectra of $^2H,^{13}C,^{15}N$-labelled R0RBR parkin bound to unlabelled pUb in the absence and presence of one equivalent unlabelled UbcH7-Ub. Chemical shift perturbations (absence of signal, average shift + 1 SD) are modelled onto the surface of R0RBR bound to pUb (PDB 5N2W). The UbcH7-Ub binding site comprises the RING1 and IBR regions (magenta). An adjacent site composed of many buried residues was also observed (cyan) that results from E2-Ub binding but does not include the E2 binding site. Sections of the tether and linker regions not visible in the crystal structure were added using Modeller (Eswar et al, 2006) so that chemical shift changes for residues not observed in crystal structures could be mapped.

C  Model of R0RBR:pUb in complex with UbcH7-Ub derived from NMR chemical shift data and HADDOCK docking (Dominguez et al, 2003). The lowest energy structure is shown, and the top 20 structures all showed excellent agreement (RMSD 0.71 ± 0.1 Å) although some minor differences were noted for the orientation of the Ub molecule. The structure shows that UbcH7-Ub binds in an open conformation. UbcH7 uses canonical E2-RING E3 interactions that include residues from two loops in the RING1 domain (L1, L2) and two loops in the E2 enzyme (L4, L7) to stabilize the interaction. Ub binds to a RING1/IBR pocket. In this model, no attempt was made to move or re-orient any of the domains in parkin.

D  Effects of ubiquitin surface mutants based on the UbcH7-Ub:R0RBR:pUb model for the ubiquitin loading and off-loading potential of pParkin$^{CH}$ (pParkin$^{C431S,H433A}$). Coomassie-stained gels (top) depict the formation of a pParkin$^{CH}$~Ub oxyester intermediate after 60 min in the absence and presence of pUb using the indicated ubiquitin species. Anti-HA blots (bottom) depict the ubiquitination of HA-tagged Miro1$^{181–579}$ by pParkin after 10 min in the absence and presence of pUb using the indicated ubiquitin species. The E24A substitution in Ub is a negative control as this residue faces away from the interface with parkin.

E  Autoubiquitination assay for parkin using ARJP and non-ARJP substitutions in parkin observed near the interface with UbcH7-Ub. Assays were done in the absence and presence of pUb. Experiments with PINK1 were done by treating parkin:pUb with PINK1 for 30 min prior to adding other reagents needed for ubiquitination. Assays were monitored using a DyLight-labelled Ub protein and measuring fluorescence at 800 nm.

chemical shift changes were in this region of the tether (Y391, R392, D394), which undergoes multiple rearrangements in crystal structures of parkin (Kumar *et al*, 2017). NMR dynamics experiments (Fig EV1; Kumar *et al*, 2015) and a structure of the isolated IBR–tether–RING2(Rcat) region (Spratt *et al*, 2013) indicate this region is very flexible and likely adopts multiple conformations in solution. We interpreted the chemical shift changes within the tether region to result from both direct E2 binding and an altering of the tether position to accommodate the E2 enzyme. The open arrangement of the E2-Ub bound to R0RBR parkin more closely resembles the interaction of UbcH5b-Ub with HOIP (Lechtenberg *et al*, 2016) than either of the structures for UbcH7-Ub with HHARI (Dove *et al*, 2017; Yuan *et al*, 2017; Fig EV4). Ub binding is governed predominantly by contacts from β1-L1-β2 (K6, L8, K11, I13-T14), the linker following helix α1 (Q31-D32) and C-terminus (L73, R74) to an R0RBR surface including β1 (P333, P335) and the C-terminus of the IBR domain (E370) and adjacent tether (V380, F381, S384, T386), RING1 helix H1 (N273) and the straightened RING1 helix H3 (R314, Y318). The Ub binding site of the IBR domain from UbcH7-Ub binding agrees well with potential ubiquitin-binding regions (UBR2, UBR3) inferred from crystallographic studies (Kumar *et al*, 2017). Further, the location of the Ub molecule provides insight into the next step of the ubiquitination process, the transfer to the RING2(Rcat) domain. The Ub conjugate is positioned such that two hydrophobic regions including the I44 patch and the C-terminus (V70, L71) of Ub are pointed away from the RING1 domain and exposed to solvent. Although not identical, the activated Ub in the HOIP/UbcH5b complex and donor Ub in the RING2L transfer complex are positioned similarly (Stieglitz *et al*, 2013; Lechtenberg *et al*, 2016). These hydrophobic regions (I44, V70, L71) are used to recruit helix $h_{L2}$ from the catalytic RING2L domain in the domain-swapped dimer structure (Lechtenberg *et al*, 2016). This suggests that a similar mechanism might exist for parkin whereby the RING2(Rcat) domain is repositioned adjacent to the hydrophobic sites in Ub. The orientation of the UbcH7-Ub conjugate in our structure poises the C-terminus of Ub for transthiolation by exposing the G76 carboxyl in the

isopeptide linkage towards the tether side of parkin. This arrangement suggests that nucleophilic attack by the catalytic C431 in the RING2(Rcat) domain would come from this direction (backside as shown in Fig 2C). The structure also shows that helix H2 in UbcH7, previously used to interact with Ub in the closed E2-Ub conformation, is exposed on the same side as the C86K-G76 linkage. Overall, the current structure shows how UbcH7 might position its Ub cargo for transfer to the catalytic cysteine (C431) of the RING2(Rcat) domain and provides clues that suggest the RING2(Rcat) domain is eventually repositioned near the UbcH7-Ub conjugate to promote Ub transfer.

In order to further test the parkin recruitment site for UbcH7-Ub observed in our models, a series of ubiquitination assays were performed. We first analysed the observed ubiquitin surface involved at the Parkin:UbcH7-Ub interface using distinct assays that monitor transfer of ubiquitin from the E2 onto parkin and subsequently onto the substrate. To assess UbcH7 mediated ubiquitin loading of parkin, we generated a pParkin RING2(Rcat) mutant (C431S + H433A, referred to as pParkin$^{CH}$) that is able to trap an E3~Ub oxyester intermediate (Spratt *et al*, 2013; Kumar *et al*, 2017). We rationalized that ubiquitin loading to pParkin$^{CH}$ would depend on the interaction of Ub in the UbcH7~Ub conjugate with pParkin. In the presence of ATP, E1, UbcH7 and wild-type ubiquitin, we observe the formation of a pParkin$^{CH}$~Ub species in both the absence and presence of pUb (Fig 2D, top). In contrast, alanine mutants of ubiquitin β1-L1-β2 residues (K6A, L8A, K11A) observed at the Ub interface with the parkin IBR domain result in defective ubiquitin transfer onto parkin even in the presence of pUb. Consequently, these ubiquitin mutants were also compromised in pParkin-mediated substrate (HA-tagged Miro1$^{181–579}$) ubiquitination (Fig 2D, bottom). The ubiquitin mutant E34A shows minimal effects suggesting the C-terminus of the long helix in Ub has a lesser role in directing the Parkin:UbcH7~Ub interface. A similar observation was made for the HOIP:UbcH5b-Ub complex (Lechtenberg *et al*, 2016). In a complementary approach, we tested the UbcH7-Ub interface with parkin by using ARJP and non-ARJP parkin variants to assess E3 ligase activity (Fig 2E). All assays were done in the absence and

presence of pUb or with PINK1 preincubated with parkin and pUb to enable phosphorylation of the Ubl domain. As expected, significant increases in ubiquitination were observed in the presence of both phosphorylation steps. ARJP variant T240R in the RING1 helix H1 shows significant decreases in activity due to disruption of interactions with F63 and P97 in UbcH7. The P335G/C337G substitution in parkin expected to disrupt one of the Zn-binding sites in the IBR domain that interacts with K11, T13 and I14 of ubiquitin also had diminished activity. The R271S substitution that is near the UbcH7 L4 loop in our structure had a minor decrease consistent with modification of that interaction. The activity of two other substitutions, N273S and R314A, had similar activities to that of the wild-type E3 ligase, likely a reflection of a weakened interaction with the Ubl facilitating its phosphorylation and stimulating ubiquitination as previously observed (Sauvé et al, 2015).

## UbcH7-Ub binding leads to a re-modelled RING0/RING2 interface in parkin

In addition to the UbcH7-Ub binding site on parkin, analysis of chemical shift perturbation experiments of $^2$H,$^{13}$C,$^{15}$N-labelled R0RBR:pUb upon addition of UbcH7-Ub reveals many changes localized at the interface between the C-terminus of the tether region (A398-T414), RING0 (S145, F146), RING1 (Q252, R256, H257) and RING2(Rcat; T415, E426, K427, N428, D464; Figs 2B, and 3A and B). In contrast, NMR experiments performed between $^2$H,$^{13}$C,$^{15}$N-labelled R0RBR:pUb and unconjugated UbcH7 revealed many of these chemical shift perturbations are absent, or present to a lesser extent (Fig EV2), suggesting the intact UbcH7-Ub conjugate is necessary to induce these additional changes. Three-dimensional structures show that most of the affected residues form a cluster anchored by W403 that follows the short helix in the tether region and is essential for packing the tether against the RING0/RING1/RING2(Rcat) core (Fig 3A and B). In the absence of Ub or Ubl phosphorylation by PINK1, a W403A parkin variant dramatically increases ubiquitination activity (Trempe et al, 2013), likely a result of a structural rearrangement near W403 and exposure of the subsequent catalytic C431 in the RING2(Rcat) domain. Therefore, we hypothesized that UbcH7-Ub binding to R0RBR:pUb might cause a similar rearrangement near the RING0/RING1/RING2 (Rcat) core in the wild-type protein as in the W403A substituted version (R0RBR$^{W403A}$). To test this, we compared the positions of resonances from a $^{15}$N-labelled R0RBR$^{W403A}$ HSQC spectrum with those from a $^{15}$N-labelled R0RBR spectrum (Appendix Fig S5). Remarkably, in the R0RBR$^{W403A}$ data many signals from residues that neighbour W403 are either undetectable or undergo significant chemical shift changes similar to those observed upon UbcH7-Ub binding to R0RBR:pUb but not directly at the E2 binding site. The similarity of the chemical shift changes and the buried nature of many of these residues suggest the RING0/RING1/RING2(Rcat) interface is re-modelled during the E2~Ub interaction with parkin.

## The UbcH7-Ub conjugate enhances pUbl domain re-binding to parkin

Upon presentation of pUb and phosphorylation of the Ubl domain of parkin, both HDX and NMR dynamics experiments show the

pUbl domain is dislodged from the autoinhibitory site against the IBR and RING1 domains. This is supported by the nearly 10-fold poorer affinity of the Ubl domain for R0RBR parkin upon phosphorylation (Kumar et al, 2015). In order to test how the UbcH7-Ub conjugate might alter the interaction of the pUbl domain with the remainder of the E3 ligase, we again used HDX mass spectrometry experiments of phosphorylated parkin bound to pUb (pParkin: pUb), this time in the presence of UbcH7-Ub. The dissociation constant for this complex is near 0.5 μM, so these experiments were done using 10 μM pParkin:pUb in the presence of non-hydrolysable UbcH7-Ub (10 μM) to achieve more than 80% saturation of pParkin:pUb with the E2-Ub conjugate during HDX measurements.

In the presence of the E2-Ub complex, the majority of parkin shows greater exchange than the phosphorylated, activated state (Fig 4A and Appendix Fig S6). In general, this indicates that a global rearrangement has occurred in the protein, stimulated by UbcH7-Ub conjugate binding. For example, the HDX data show significant increases in exchange for the RING0 and RING2(Rcat) domains. This observation is incongruous with current structures of parkin, and our HDX data for pParkin:pUb, that show most of these domains are protected from solvent and not exposed. In contrast, these regions correspond closely to the RING0/RING1/RING2(Rcat) interface with the tether region suggested to be re-modelled based on NMR chemical shift perturbation experiments (Fig 3A and B). The increase in deuteration upon UbcH7-Ub binding indicates this interface is becoming much looser with a loss of both hydrophobic and hydrogen bonding interactions. One possible interpretation of this result is that the RING2(Rcat) domain is displaced from the interface exposing its surface, the RING0 interface with RING2(Rcat) and the C-terminus of the tether (Fig 4B).

Coincident with the changes in HDX for the RING0 and RING2 (Rcat) domains, numerous increases are noted near the pUb interface with the RING0/RING1 cleft and extending towards the IBR domain. These regions undergo multiple rearrangements in structure and orientation upon pUb and Ubl binding (Kumar et al, 2015; Wauer et al, 2015; Kumar et al, 2017). The increase in HDX could indicate further rearrangement occurs upon UbcH7-Ub binding leading to a more extended structure. The HDX experiments also show that the pUbl domain remains exposed to solvent. One exception is the C-terminus of the pUbl domain (D62-M80) that is significantly more protected. This region contains the phosphorylated S65 residue suggested to interact with three basic residues (K161, R163 and K211) in the RING0 domain that show attenuated parkin autoubiquitination when substituted (Wauer et al, 2015). Two of these sites are locations of ARJP substitutions (K161N, K211R, K211N). A minority of parkin crystal structures also show a bound sulphate ion in this area. Although we did not observe a decrease in HDX in this region of the RING0 domain, the decreased exchange in the N-terminus of the pUbl domain could be consistent with increased binding to the RING0 domain stimulated by the UbcH7-Ub conjugate.

To test how UbcH7-Ub binding might influence the potential relocation of the pUbl domain, we examined a series of $^1$H–$^{15}$N HSQC spectra for $^{15}$N-labelled phosphorylated parkin in complex with unlabelled pUb and compared this to the isolated pUbl domain. We then compared spectra of $^{15}$N-labelled pUbl in the absence and presence of R0RBR:pUb and UbcH7-Ub. In full-length

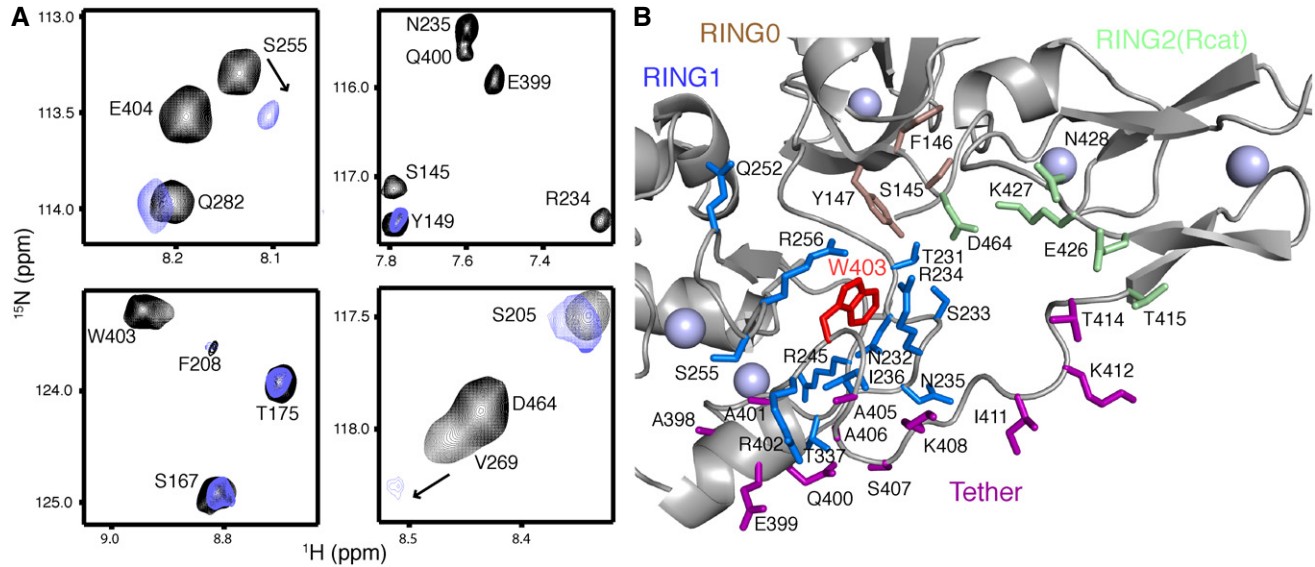

**Figure 3. UbcH7-Ub binding induces re-modelling of the RING0/RING2(Rcat) interface.**

A   Selected regions from $^1H$–$^{15}N$ TROSY spectra of $^2H,^{13}C,^{15}N$-labelled R0RBR parkin bound to unlabelled pUb in the absence (black contours) and presence (blue contours) of one equivalent invisible $^2H$-labelled UbcH7-Ub. A large number of signals for residues including S145, N235, S255, E399, Q400, W403, E404 and D464 either shift or are absent in spectra with UbcH7-Ub.

B   Details of the re-modelled region in parkin observed upon UbcH7-Ub binding. The figure shows the side chains of residues in the RING0 (brown), RING1 (blue), tether (purple and RING2(Rcat; pale green) interface that surround W403 (red) and are most affected by UbcH7-Ub binding.

phosphorylated parkin, the $^1H$–$^{15}N$ HSQC spectrum is very complicated due to a large number of signals that are visible from the highly flexible linker and tether regions in the protein (Fig 4C). However, nearly all signals are visible from the pUbl domain in the full-length protein when a short relaxation delay is used in the HSQC experiment to suppress broad signals arising from the compact, folded R0RBR region (see also fig S9 in Aguirre *et al*, 2017). When compared to the $^1H$–$^{15}N$ HSQC spectrum of the isolated pUbl domain (Fig 4C), it is clear that very small chemical shift changes and decreases in intensities occur for residues pSer65, D62, Q63 and Q64, consistent with a weak interaction of the pUbl with the remainder of the protein. However, many other signals from the pUbl are hardly attenuated (Aguirre *et al*, 2017). These observations are consistent with our HDX experiments, which indicate the pUbl domain spends most of its time dissociated from the remainder of the protein in the absence of an E2-Ub conjugate (Fig 1D).

When the $^1H$–$^{15}N$ HSQC spectrum of $^{15}N$ pUbl is compared to that with added R0RBR:pUb, we observe no significant chemical shift changes (Fig 4D). This supports previous observations of a weak interaction for the pUbl domain with R0RBR parkin *in trans* (Kumar *et al*, 2015; Sauvé *et al*, 2015). Upon addition of UbcH7-Ub, we observe small but measurable changes for the positions of several signals including pSer65, D62, Q63 and Q64 (Fig 4D). These chemical shift changes occur on the fast exchange timescale consistent with weak binding of the pUbl domain to R0RBR parkin that is accentuated upon UbcH7-Ub recruitment by the E3 ligase. This indicates that the pUbl domain undergoes a weak interaction with the remainder of the parkin protein that is stimulated through binding of the UbcH7-Ub conjugate.

## UbcH7-Ub and pUbl work together to modulate C431 reactivity

Our NMR experiments using R0RBR parkin allowed us to determine the UbcH7~Ub binding site within parkin as one requirement for ubiquitination and have identified how UbcH7-Ub binding might re-model the RING0/RING1/RING2(Rcat) interface with the tether. However, in the absence of the pUbl domain these experiments are deficient in establishing how the pUbl might work with an E2~Ub conjugate to achieve optimal activity. In particular, all three-dimensional structures to date have been unable to show how the reactivity of the catalytic C431 residue in the RING2(Rcat) domain might be altered upon phosphorylation of the Ubl domain and addition of the E2~Ub conjugate. In order to assess this, we examined the ability of full-length parkin and R0RBR parkin to form a non-hydrolysable ubiquitin adduct using ubiquitin~vinyl sulphone (UbVS). Although smaller probes to test E3 ligase activity are available (Pao *et al*, 2016), we used UbVS to mimic the parkin-Ub product that would be expected following Ub transfer from the UbcH7 enzyme. Using different combinations of parkin, phosphorylated parkin, phospho-ubiquitin and UbcH7-Ub, this approach tested the reactivity, and hence accessibility, of the catalytic C431 in the RING2(Rcat) domain during each step of the parkin activation cycle (Fig 5).

As expected, autoinhibited and pUb-bound parkin or R0RBR showed minimal reaction with UbVS (Fig 5 and Appendix Fig S7) in agreement with previous three-dimensional structures and reactivity profiles that indicate the catalytic C431 is mostly occluded by neighbouring RING0 domain interactions (Riley *et al*, 2013; Wauer & Komander, 2013). Remarkably, phosphorylated parkin activated by pUb shows rapid product formation with UbVS (Fig 5A and B),

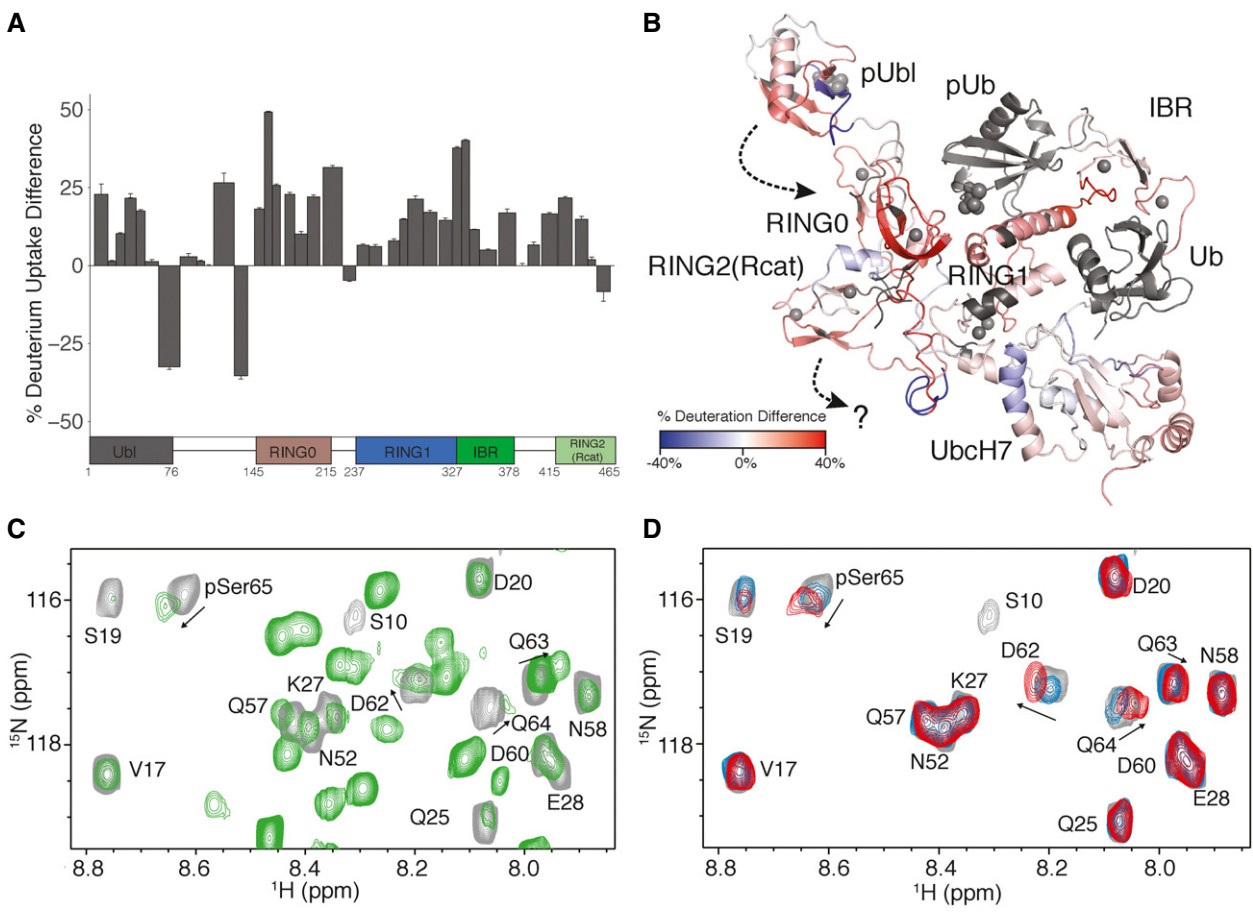

**Figure 4.   Interaction of the pUbl domain is facilitated by UbcH7-Ub recruitment.**

A   Differences in % deuterium uptake between the pParkin:pUb and pParkin:pUb in complex with UbcH7-Ub states plotted at the one-minute time point. The width of the bars shown represents the length of the peptides observed and measured, while the heights represent the relative difference in hydrogen–deuterium exchange between the two species. Error bars represent standard deviation above the average for triplicate measurements. Positive bars indicate a greater deuterium uptake in the pParkin:pUb:UbcH7-Ub state compared to the pParkin:pUb state, signifying a greater exposure to solvent, while a negative change indicates the complex state is more protected. The linear domain structure is shown below the data to indicate the location of peptides.

B   Relative HDX differences from (A) are mapped to the structure of pParkin:pUb and UbcH7-Ub complex. Regions where HDX is faster in the pParkin:pUb:UbcH7-Ub complex at the one-minute time point are indicated in red, while those regions in blue exchange more slowly using a similar the gradient as Fig 1. Based on HDX and NMR experiments (C, D), the pUbl domain is shown oriented towards the RING0 domain (dashed arrow, top). The increased HDX for RING2(Rcat) and RING0 suggests the RING2(Rcat) domain has a weaker interaction with the RING0/RING1 domains upon UbcH7~Ub binding (dashed arrow, bottom). Regions shown in grey are peptides that were not mapped.

C   $^1$H–$^{15}$N HSQC spectra of full-length $^{15}$N-labelled pParkin:pUb (green contours) compared to that for the isolated $^{15}$N-labelled pUbl domain (grey contours). The spectrum of the full-length pParkin:pUb complex was collected using a $^1$H–$^{15}$N CPMG, T2-filtered HSQC (Aguirre *et al*, 2017) to show only the more mobile regions of pParkin. In the spectrum of full-length pParkin, many additional resonances arising from flexible loops in the protein are observed and are not labelled for clarity. Residues that experience small chemical shift changes are indicated by arrows.

D   $^1$H–$^{15}$N HSQC spectra of $^{15}$N-labelled pUbl (grey contours) and in the presence of an equimolar amount of unlabelled R0RBR:pUb (blue contours) show little change. Upon addition of unlabelled UbcH7-Ub (red contours), small changes in chemical shifts of pSer65, D62, Q63 and Q64 are observed as indicated by arrows.

visible even after 1 min (not shown). A similar reaction with R0RBR:pUb when the parkin pUbl domain is added *in trans* shows little modification with UbVS even after 60 min (Fig 5C and D). The most logical explanation for these observations is that the pUbl domain is facilitating access of the UbVS probe to the catalytic C431 site in the RING2(Rcat) domain by binding to another region in parkin. Since the pUbl interaction exhibits a weaker affinity *in trans*, the effect of the UbVS probe is much lower than observed for the intact protein. Interestingly, introduction of UbcH7-Ub to either the

pParkin:pUb sample (Fig 5A and B) or the pUbl:R0RBR:pUb complex (Fig 5C and D) leads to opposite results. In the full-length protein, we observe a reproducible lower conversion rate in the presence of the E2-Ub conjugate than in its absence (Fig 5A and B). In the pUbl:R0RBR:pUb sample, the reactivity to UbVS is enhanced in the presence of UbcH7-Ub (Fig 5C and D). At a minimum, this shows that the pUbl domain and E2-Ub act synergistically to alter the reactivity of the catalytic C431 residue in the RING2(Rcat) domain. Introducing the "activating" W403A mutation did not

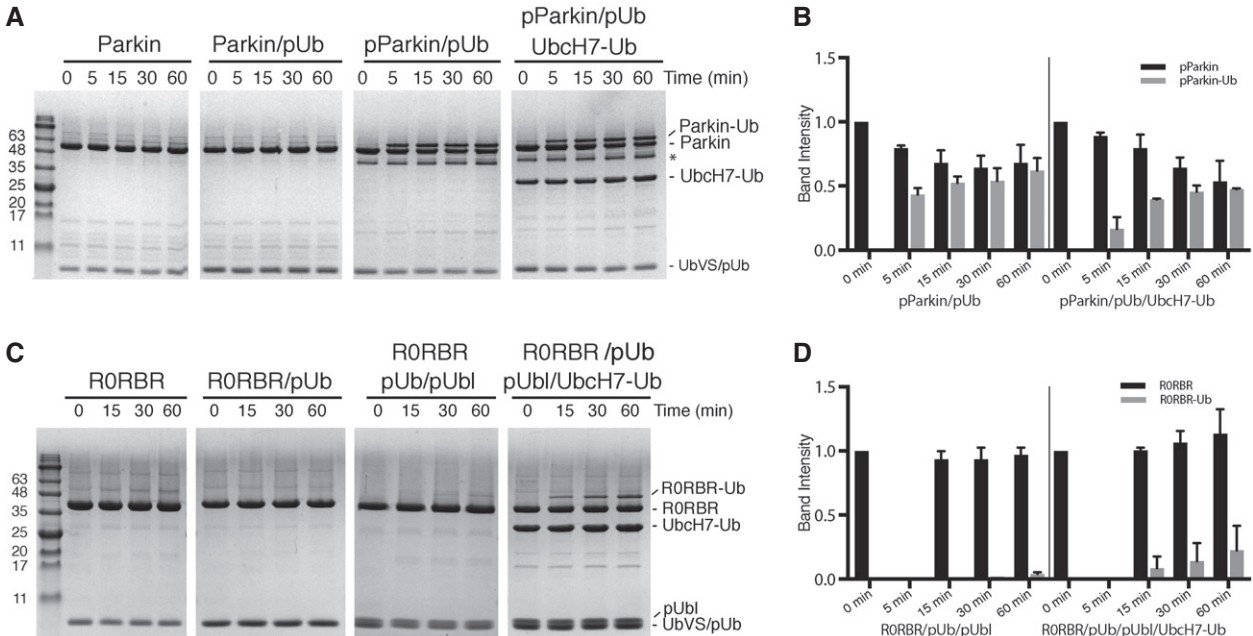

**Figure 5. The pUbl domain and UbcH7-Ub synergistically modulate catalytic C431 reactivity.**

A   Exposure of the catalytic Cys431 is parkin as assessed by reaction with a UbVS probe. The different stages of activation using parkin, parkin:pUb, pParkin:pUb and pParkin:pUb in combination with an isopeptide-linked UbcH7-Ub conjugate are indicated above each gel. Following addition of UbVS, samples were taken at the times indicated (0–10 min) and visualized by SDS–PAGE.

B   Relative percentages of pParkin and the pParkin-Ub adduct as a function of time. Intensity percentages were calculated as a function of total intensity of pParkin-Ub, parkin and UbVS/pUb bands. Error bars represent standard deviation from the average for duplicate measurements.

C   Exposure of the catalytic Cys431 is R0RBR parkin as assessed by reaction with a UbVS probe. The different stages of activation using R0RBR, R0RBR:pUb, R0RBR:pUb:pUbl and R0RBR:pUb:pUbl in combination with an isopeptide-linked UbcH7-Ub conjugate are indicated above each gel. Following addition of UbVS, samples were taken at the times indicated (0–60 min) and visualized by SDS–PAGE.

D   Relative percentages of R0RBR and the R0RBR-Ub adduct as a function of time. Intensity percentages were calculated as a function of total intensity of the R0RBR-Ub, R0RBR and UbVS/pUb/pUbl bands.

significantly increase reactivity with the UbVS probe, suggesting the re-modelling observed in our NMR experiments may not directly expose parkin's catalytic Cys431 (Appendix Fig S7). In agreement with our NMR data (Fig 4), the E2-Ub conjugate has the ability to re-model the RING0/RING2(Rcat) interface and increase binding of the pUbl domain *in trans* that results in increased reactivity of the catalytic C431 residue. Meanwhile, the small decrease in C431 accessibility in the pParkin:pUb complex in the presence UbcH7-Ub suggests a further conformational change occurs due to E2~Ub binding that decreases the availability of the catalytic site. This is in agreement with our HDX experiments that suggest a reorganization of the RING2(Rcat) domain (Fig 4).

## Discussion

Although parkin was originally thought to be a constitutively active enzyme, it is now known to regulate its activity through intramolecular domain–domain interactions and binding to effectors (Chaugule *et al*, 2011; Kumar *et al*, 2015; Sauvé *et al*, 2015). In particular, PINK1 regulates parkin activity through phosphorylation of the Ubl domain, and of ubiquitin itself (Kumar *et al*, 2015; Sauvé *et al*, 2015; Wauer *et al*, 2015; Kumar *et al*, 2017), which then acts as an

effector. Multiple structures of autoinhibited parkin reveal that the E2 binding site is blocked; however, static crystal structures have shown that binding to pUb does not render the proposed E2 binding site, nor the phosphorylation site in the Ubl domain accessible (Fig 6A). Comparison of crystal structures in the absence/presence of the Ubl domain reveals reorganization of residues between the RING0/RING1 interface and multiple arrangements of the IBR domain, shown to be flexible by previous NMR dynamics experiments (Kumar *et al*, 2015). Though pUb binding to parkin decreases the affinity of R0RBR for the Ubl domain, it is not sufficient to dislodge the Ubl domain in the crystal (Kumar *et al*, 2015, 2017). Rather, this step optimizes parkin for Ubl phosphorylation and E2~Ub engagement (Fig 6B; Kumar *et al*, 2015; Ordureau *et al*, 2015). It is well established that phosphorylation of the Ubl domain following pUb recruitment significantly increases its ubiquitination activity (Kondapalli *et al*, 2012; Shiba-Fukushima *et al*, 2012; Kane *et al*, 2014; Kazlauskaite *et al*, 2014; Koyano *et al*, 2014). Current NMR dynamics analysis of R0RBR parkin:pUb as a proxy for this state shows the IBR domain is considerably less mobile due to engagement with pUb although a large stretch of the tether region remains mobile, an observation not obvious from current crystal structures. In full-length phosphorylated parkin bound to pUb ("Partial Activation"; Fig 6C), current and other HDX experiments

(Gladkova *et al*, 2018; Sauvé *et al*, 2018) show that the pUbl domain is nearly completely exposed to solvent in this state compared to its position in autoinhibited parkin where it is surrounded by the RING1, IBR and tether regions. This interpretation agrees with NMR relaxation data, analytical centrifugation experiments and computational work that conclude the pUbl domain samples a large conformational space that includes weak interaction with other regions of parkin (Caulfield *et al*, 2014; Aguirre *et al*, 2017; Fig 6C). While this manuscript was under review, two crystal structures of partially activated parkin appeared, which showed the pUbl domain has the ability to bind to the RING0 domain (Gladkova *et al*, 2018; Sauvé *et al*, 2018) comprised of a previously identified (Wauer & Komander, 2013) basic patch (K161, R163, K211) that included two ARJP substitutions. Mutation of these residues renders parkin unreactive with an E2-based activity probe, consistent with a requirement for pUbl interaction (Pao *et al*, 2016). Yet other experiments show parkin retains appreciable ubiquitination ability in the absence of its Ubl domain (Chaugule *et al*, 2011; Kazlauskaite *et al*, 2014; Kumar *et al*, 2015), suggesting phosphorylation and the Ubl domain itself are less important. Our NMR data of full-length pParkin:pUb (Fig 5) indicate the native interaction of the pUbl domain with the RING0 domain is weak in this state in agreement with previous affinity experiments (Kumar *et al*, 2015; Sauvé *et al*, 2015). Consistent with this, it was necessary to remove the entire RING2(Rcat) domain and the tether region from parkin in recent structures in order to capture the bound pUbl state (Gladkova *et al*, 2018; Sauvé *et al*, 2018), suggesting removal of the RING2 (Rcat) domain may enhance the pUbl interaction through removal of steric hindrance.

One of the main outcomes of pUb recruitment and subsequent phosphorylation of the Ubl domain is to unmask the RING1 binding site for the E2 conjugating enzyme and rearrangement of the IBR domain to engage the conjugated Ub molecule (Kumar *et al*, 2015; Kumar *et al*, 2017). This interaction with UbcH7-Ub shows that the donor Ub is in the open conformation favoured by RBRs (Dove *et al*,

2016). The interaction site of UbcH7 with the RING1 domain in our structure is in agreement with previous crystal structures of HOIP with UbcH5b-Ub (Lechtenberg *et al*, 2016) and HHARI with UbcH7-Ub (Dove *et al*, 2017; Yuan *et al*, 2017) although minor orientation differences occur. This likely arises due to differences in the L2 loop regions of the RBR E3 ligases as previously noted (Spratt *et al*, 2014; Dove *et al*, 2016), and in particular differences in the linchpin residue that directs the E2~Ub conjugate to its open state during interaction (Dove *et al*, 2017). The position of the UbcH7 enzyme (in UbcH7-Ub) with parkin is also nearly identical to a recent structure of truncated parkin in complex with an unconjugated E2 enzyme (Sauvé *et al*, 2018). This indicates the donor Ub does not play a major role in directing the E2 binding in agreement with its poor binding affinity for parkin on its own.

Our NMR data indicate that UbcH7-Ub binding to partially activated parkin re-models the hydrophobic cluster involving W403 in the tether region at the junction of the RING0/RING1/RING2(Rcat) domains. This is supported by substitution of W403 that produces NMR chemical shift changes analogous to those for binding of UbcH7-Ub. Further, experiments using an E2-based activity probe show a W403A substitution can partially recapitulate catalytic cysteine labelling of parkin even in the absence of Ubl domain phosphorylation (Pao *et al*, 2016).

Together these observations suggest that it is the E2~Ub binding step that induces a conformational change in the W403 cluster (Fig 6D), rather than W403 being re-modelled to allow E2~Ub engagement, that facilitates the interaction of the pUbl domain with the RING0 domain of parkin. Although this step could not be shown in recent crystal structures that lacked the RING2(Rcat) domain and tether regions, our HDX data show that both phosphorylation and E2-Ub binding together lead to nearly complete exposure of the RING2(Rcat) domain. The interplay between pUbl and E2~Ub binding to modulate ubiquitination efficiency is also borne out from the reactivity of the catalytic cysteine (C431) in RING2(Rcat) to react with a ubiquitin probe. In R0RBR parkin, reactivity required both

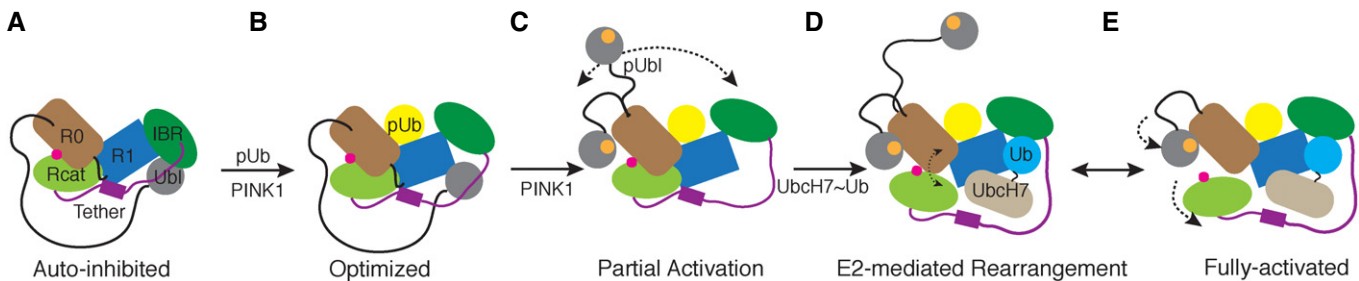

**Figure 6. Model of parkin activation by combined pUbl and E2~Ub interactions.**

A  Autoinhibited state whereby the Ubl domain masks the E2 binding site on the RING1 (R1) domain as described by Chaugule *et al* (2011) and supported through crystallographic studies.

B  Optimization step controlled by PINK1 phosphorylation of ubiquitin (Ub) and subsequent binding of pUb to the RING1/IBR interface resulting reorganization of the RING0/RING1 interface and movement of the IBR domain. Steps (A) and (B) have been previously described.

C  PINK1 phosphorylation of the Ubl domain causes its dissociation from the RING1 binding site allowing it to sample a large conformational space in solution including weak binding to the RING0 domain favouring an uncovering of the catalytic cysteine C431.

D  E2~Ub binding to the RING1/IBR domains making use of a cryptic ubiquitin binding site uncovered through dissociation of the pUbl domain. Re-modelling of the RING0/RING1/RING2(Rcat) interface with the tether region occurs based on NMR chemical shift data and HDX experiments.

E  Fully activated state of parkin utilizes synergistic binding of pUbl domain to the RING0 domain observed in crystal structures (Gladkova *et al*, 2018; Sauvé *et al*, 2018) and E2~Ub binding that re-models the RING0/RING1/RING2(Rcat) interface to maximize accessibility to the catalytic cysteine.

pUbl and E2-Ub addition for adduct formation. In similar NMR experiments, addition of E2-Ub was required to observe binding of pUbl to R0RBR parkin. The fast exchange observed in current NMR experiments between the pUbl domain and the RING0 site observed in crystal structures (Gladkova *et al*, 2018; Sauvé *et al*, 2018) suggests that this interaction is short-lived. We suggest that upon E2~Ub recruitment, the RING2(Rcat) domain is dislodged to provide a more optimal binding site for the pUbl domain. Indeed, one recent structure shows that an additional segment of the linker that precedes the RING0 domain takes the place of the RING2(Rcat) domain (Gladkova *et al*, 2018), perhaps providing an extra level of regulation. Taken together, we suggest that pUbl binding alone to the RING0 domain is not sufficient to drive the conformational changes required and that E2~Ub binding is necessary to re-model the W403 region for full activation (Fig 6D and E). These events show that both E2~Ub binding and pUbl recruitment to the RING0 domain act synergistically to propagate the ubiquitination reaction.

Recruitment of the UbcH7-Ub conjugate to pParkin:pUb appears to cause a significant conformational change based on HDX and NMR chemical shift experiments. An unresolved question then is: What is the conformational change that occurs that allows ubiquitin transfer from the E2 conjugating enzyme to the RING2(Rcat) domain? It is tempting to consider that the RING2(Rcat) domain repositions itself nearby the open E2~Ub conjugate to accept the donor ubiquitin. In the absence of data to directly show this, several lines of evidence suggest this is a possibility. For example, our HDX data show that although the RING2(Rcat) domain is exposed in the presence of UbcH7-Ub, its extreme C-terminus (E452-W462) is more protected compared to that in the absence of E2-Ub. This portion of parkin has been suggested to form important interactions with the E2~Ub conjugate based on the similarity of hydrophobic residues with HOIP and the position of the swapped RING2L domain in HOIP that neighbours the UbcH5b-Ub conjugate (Lechtenberg *et al*, 2016). UbVS experiments show that the reactivity of C431 in RING2(Rcat) is slightly diminished in the presence of UbcH7-Ub (compared to its absence in pParkin:pUb), suggesting possible rearrangement and protection might impede its reactivity in this assay. Further, the region immediately preceding the RING2(Rcat) domain contains a ubiquitin-binding motif (Chaugule *et al*, 2011) that harbours at least one ARJP substitution (T415N) that impairs all ubiquitination activity (Matsuda *et al*, 2010). A similar ubiquitin-binding motif has been observed for HHARI, and substitutions in this region show significant decreases in Ub affinity and ubiquitination activity (Dove *et al*, 2016). Although unstructured in parkin, the corresponding region in HOIP forms a short α-helix (helix $h_{L2}$) that interacts with the I44 patch of the donor Ub in the UbcH5b-Ub complex. These observations hint at a catalytic complex where interactions between both components of the UbcH7~Ub conjugate are important for recruitment of the tether and RING2(Rcat) regions. Alternatively, multiple studies have suggested that the ubiquitin-transfer complex is more complicated and requires co-operation between multiple parkin and E2~Ub molecules (Lazarou *et al*, 2013; Kumar *et al*, 2017) to facilitate ubiquitin transfer. An attractive feature of this model, which also utilizes alternate ubiquitin binding sites, is that it provides a framework for the processivity of autoubiquitination observed for parkin. With this in mind, it is worth noting that recent crystal structures lack structural resolution of several linkers in parkin that preclude identification of intra- vs. intermolecular interactions of

the pUbl or E2 enzyme with the RING0 or RING1 domains, respectively (Gladkova *et al*, 2018; Sauvé *et al*, 2018). By the same note, our NMR-based model of UbcH7-Ub in complex with R0RBR:pUb is unable to identify how the tether and RING2(Rcat) might be rearranged upon UbcH7-Ub and pUbl engagement. Nevertheless, our study shows that synergy exists between these two effectors to fully control parkin activity. Understanding the next steps in detail will be essential to target this important enzyme for modulation during the pathogenesis of Parkinson's disease.

## Materials and Methods

### Protein constructs and purification

Full-length human parkin (1–465), Ubl (1–76), R0RBR (141–465), *Drosophila melanogaster* RING2 (410–482) and other parkin variants were expressed and purified as described previously (Chaugule *et al*, 2011; Spratt *et al*, 2013). Briefly, His-smt3-parkin constructs were expressed in BL21(DE3) cells at 37°C to an $OD_{600}$ of 0.8. Expression was induced at 16°C with 25 μM IPTG for parkin, 0.1 mM IPTG for R0RBR and 0.5 mM IPTG for Ubl or RING2 for 18 h. All growths, except the Ubl domain, were supplemented with 0.5 mM $ZnCl_2$. Purification utilized an initial HisTrap FF column followed by Ulp1 cleavage at 4°C, a second HisTrap FF column and final Superdex 75 10/300 size-exclusion chromatography. Selectively $^2H,^{13}C,^{15}N$-labelled R0RBR:pUb or R0RBR and selectively $^2H,^{13}C,^{15}N$ or $^2H,^{15}N$-labelled Ub, UbcH7 or UbcH7-Ub were expressed and purified as previously described (Kumar *et al*, 2015).

His-tagged Uba1 was expressed in BL21(DE3)CodonPlus-RIL cells at 37°C to an $OD_{600}$ of 0.8. Expression was induced with 0.5 mM IPTG at 18°C for 12 h. His-tagged Uba1 was purified on a HisTrap FF column by washing with a buffer containing 50 mM Tris, 200 mM NaCl, 250 μM TCEP and 25 mM imidazole (pH 8.0) and then washing with 14% of elution buffer that contained 250 mM imidazole. The His-tagged Uba1 was then eluted with 100% elution buffer and stored in aliquots at −80°C for ubiquitination assays.

His-TEV-tagged human $UbcH7^{C17S,C86K,C137S}$ was expressed in BL21(DE3)CodonPlus-RIL cells at 37°C to an $OD_{600}$ of 0.8, and expression was induced with 1 mM IPTG at 30°C for 18 h. UbcH7 was purified on a HisTrap FF column, cleaved at 4°C overnight and purified on a second HisTrap FF column. His-tagged Ub was expressed in BL21(DE3)CodonPlus-RIL cells. Complexes that contained 1:1 R0RBR:pUb were formed using a 1.5-fold excess pUb compared to R0RBR in a buffer containing 20 mM Tris, 75 mM NaCl and 250 μM TCEP (pH 8). The 1:1 complex mixture was purified on a Superdex 75 10/300 size-exclusion column to ensure excess pUb was not present in samples for NMR studies.

Glutathione S-transferase (GST)-HA-tagged human Miro1 (residues 181–592) was expressed in BL21(DE3) cells purified using standard protocols (Kumar *et al*, 2015). The GST tag was removed using GST-3C protease, and the cleaved material was further purified on a Superdex 200 Increase 10/300 size-exclusion column.

### Protein phosphorylation

Phosphorylation of Ub, Ubl and parkin was done using purified *Pediculus humanus* PINK1 (126–575) as described previously

(Kumar *et al*, 2015; Aguirre *et al*, 2017). For pUb and pUbl (1–76), typically 10 μM PINK1 was sufficient to stoichiometrically phosphorylate either 100 μM Ub or Ubl in 1 h at 24°C. For parkin, typically 75 μM PINK1 was sufficient to phosphorylate 150 μM parkin for 3.5 h at room temperature. Reactions were visualized by Phos-tag gel. PINK1 was removed using a GSTrap FF column. Phosphorylated proteins were purified using a Superdex 75 10/300 size-exclusion column and confirmed by mass spectrometry.

## Synthesis of UbcH7-Ub isopeptide-linked conjugate

UbcH7-Ub isopeptide-linked conjugate was synthesized using an optimized version of the protocol of Plechanovová *et al* (2012). Briefly, 200 μM His-tagged Ub, 400 μM UbcH7$^{C17S/C86K/C137S}$, 25 μM non-cleavable His-tagged Uba1 and 10 mM Mg$^{2+}$/ATP were incubated together in a buffer containing 50 mM CHES and 150 mM NaCl (pH 9.0) at 37°C for 6–16 h to form approximately 60% UbcH7-Ub isopeptide-linked conjugate based on SDS–PAGE analysis. The resulting mixture was passed through a HisTrap FF column to eliminate unconjugated UbcH7. The eluted His-tagged proteins were TEV-cleaved overnight at 4°C. The mixture was purified on a second HisTrap FF column to eliminate non-cleavable His-tagged Uba1. The remaining UbcH7-Ub was separated from unreacted Ub using a HiLoad Superdex 16/60 size-exclusion column.

## NMR experiments

All NMR data were collected at 25°C on a Varian Inova 600-MHz NMR spectrometer equipped with a triple-resonance cryogenic probe and z-field gradients. Samples were prepared in a buffer containing 25 mM HEPES, 50 mM NaCl and 500 μM TCEP (pH 7.0) with 10% D$_2$O (v/v) using DSS as an internal reference and imidazole to monitor pH. Backbone assignments of R0RBR parkin (Kumar *et al*, 2015) and the pUbl domain (Aguirre *et al*, 2017) were completed using standard triple-resonance methods as previously reported. $^1$H–$^{15}$N HSQC spectra were collected in TROSY mode (Pervushin *et al*, 1997) to follow amide backbone chemical shift perturbations. $^1$H–$^{13}$C HMQC spectra (Tugarinov *et al*, 2004) were collected to monitor chemical shifts of Ub side chain methyl groups. $^1$H–$^{15}$N TROSY spectra were collected using different combinations of $^2$H,$^{13}$C,$^{15}$N-labelled and $^2$H,$^{15}$N-labelled R0RBR:pUb or R0RBR with $^2$H,$^{13}$C,$^{15}$N or $^2$H,$^{15}$N-labelled pUbl, Ub, UbcH7 or UbcH7-Ub. For the $^{15}$N-labelled pUbl binding experiments, $^1$H–$^{15}$N HSQC experiments were collected to monitor pUbl chemical shift changes with addition of equimolar amounts of unlabelled R0RBR:pUb and UbcH7-Ub. Chemical shift perturbation measurements for amide backbone resonances were calculated using $\Delta\delta = ((\Delta\delta H)^2 + (\Delta\delta N/5)^2)^{0.5}$ and for side chain methyl groups using $\Delta\delta = ((\Delta\delta H)^2 + (\Delta\delta C/3.3)^2)^{0.5}$. All data were processed using 60°-shifted cosine bell-weighting functions using NMRPipe and NMRDraw (Delaglio *et al*, 1995) and were analysed using NMRViewJ (Johnson & Blevins, 1994).

## Ubiquitin~Vinyl sulphone reactions

Individually purified proteins were acquired as described above. The final concentration of each component was 10 μM in a final volume of 45 μl [R0RBR and RING2(Rcat)] or 55 μl (parkin) in

50 mM HEPES and 50 mM NaCl (pH 8.0). Time started when the UbVS (Boston Biochem) was added to the reaction at 37°C. 10 μl was removed at each time point, and the reaction was quenched with 3× SDS sample buffer. 16.5% SDS–PAGE gels were run and stained with Coomassie Blue. Gels were imaged on a Bio-Rad ChemiDoc XRS$^+$. Band intensities of parkin-Ub (R0RBR-Ub), parkin (R0RBR) and UbVS/pUb/pUbl were measured using ImageJ software (Schneider *et al*, 2012). The per cent contribution per band was calculated from the normalized intensity in each lane.

## Ubiquitination assays

These assays were monitored by fluorescence using Ub containing an N-terminal cysteine residue linked to DyLight 800 Maleimide (Ub$^{800}$; Thermo Fisher Scientific) as previously described (Kumar *et al*, 2015; Aguirre *et al*, 2018). All reactions were performed at 37°C and contained purified 1 μM wild type or substituted parkin, 0.5 μM UbcH7, 0.1 μM Uba1, 4 μM Ub and 0.5 μM Ub$^{800}$ in 5 mM MgATP, 50 mM HEPES (pH 7.5). pUb was added to 0.5 μM when needed. To induce *in situ* phosphorylation, 0.01 μM of purified GST-PINK1 was added to the parkin/pUb/ATP samples 30 min before initiating ubiquitination. The ubiquitination reactions were quenched with 3×SDS sample buffer and 1 M DTT. 4–12% Bis-Tris gradient gels (Thermo Fisher Scientific) were used with MES running buffer (250 mM MES, 250 mM Tris, 0.5% SDS and 5 mM EDTA, pH 7.3). Fluorescence intensity at 700 and 800 nm was measured using an Odyssey Imaging System (LI-COR).

HA-Miro1 ubiquitination reactions were performed at 30°C for 10 min in a buffer containing 50 mM Tris (pH 7.5), 100 mM NaCl, 5 mM MgCl$_2$, 1 mM TCEP and 5% (v/v) glycerol. Reactions contained 25 nm E1, 250 nM UbcH7, 250 nM pParkin, 10 μM Ub (wild type or mutant), 5 μM HA-Miro1 and 5 mM ATP in final reaction volume of 10 μl. Non-activatable pUb-6His (2.5 μM) was used as an allosteric activator where indicated. Reactions were terminated using NuPAGE LDS Sample Buffer (Invitrogen), resolved on 4–12% Bis-Tris gradient gels (Thermo Fisher Scientific) and transferred onto nitrocellulose membranes using iBlot Gel Transfer Device (Invitrogen). Membranes were subjected to immunoblotting using anti-HA mouse monoclonal primary antibody (901515, Bio Legend, 1/5,000 dilution) and fluorescent-labelled secondary antibody (926-32213, Li-COR, 1/10,000 dilution). Blots were visualized using Li-COR Odyssey Infrared Imaging System.

## Parkin ubiquitin loading

Reactions monitoring pParkin-ubiquitin oxyester formation were performed at 30°C for 60 min in a buffer containing 50 mM Tris (pH 7.5), 100 mM NaCl, 5 mM MgCl$_2$, 1 mM TCEP and 5% (v/v) glycerol. Reactions contained 100 nM E1, 2.5 μM UbcH7, 2.5 μM pParkin$^{C431S + H433A}$, 10 μM Ub (wild type or mutant) and 5 mM ATP in final reaction volume of 10 μl. Non-activatable pUb-6His (2.5 μM) was included as an allosteric activator where indicated. Reactions were stopped using NuPAGE LDS Sample Buffer (Invitrogen) that contained reducing agents and boiled for 5 min. The samples were resolved on 4–12% Bis-Tris gradient gels (Thermo Fisher Scientific) and analysed by Coomassie staining.

## Hydrogen–deuterium exchange mass spectrometry

Deuteration of proteins occurred at $20 \pm 1°C$ in 90% $D_2O$ and 10% $H_2O$ with 50 mM HEPES, 100 mM NaCl and 250 μM TCEP at pH 7.0. Parkin and pParkin:pUb sample concentrations were 1 μM. UbcH7-Ub conjugate and pParkin:pUb concentrations were both 10 μM and were diluted to 1 μM in the above $H_2O$ buffer at pH 2.3. 100-μl aliquots were removed at time points between 15 s and 10 min following deuteration. Aliquots were quenched with ice-chilled 10% HCl in $H_2O$ to reach a pH of 2.3 and then flash-frozen in liquid nitrogen. Zero time point ($m_0$) controls were created by adding ice-chilled $D_2O$ to an ice-chilled protein sample under quench conditions (pH 2.3) and flash-frozen. Fully exchanged controls ($m_{100}$) were also created by exposing the proteins to $D_2O$ at pH 2.3 and heated to 70°C with a water bath for 8 h. These samples were then flash-frozen in liquid nitrogen. Aliquots were thawed to approximately 0°C and injected into a Waters HDX nanoACQUITY HPLC system. Online digestion of the proteins was performed with a POROS pepsin column at 15°C. The resulting peptides were trapped and analysed on a Waters BEH C18 column at 0°C using a water/ acetonitrile with 0.1% formic acid gradient at 40 μl/min. The peptide masses were measured using a Waters Synapt G2 Q-TOF mass spectrometer. Peptides were identified through MS/MS. The resulting peptides were analysed with Waters DynamX 3.0. Deuteration is expressed here as per cent deuteration uptake where $m_t$ is the centroid mass at time $t$ and $m_0$ and $m_{100}$ are described above according to the following equation:

$$\% \text{ D uptake} = \frac{m_t - m_0}{m_{100} - m_0} * 100$$

## Model determinations for UbcH7-Ub binding to R0RBR:pUb

Interacting residues were identified from NMR experiments (described above) and defined as those amides that shifted greater than the average + one standard deviation and residues that broadened/ shifted and could not be identified in the bound state (Fig EV2 and Appendix Fig S3). These residues were filtered for those that had > 20% side chain accessible surface area in each starting set of coordinates (below). Passive residues were defined according to the HADDOCK protocol (Dominguez *et al*, 2003; Bonvin *et al*, 2018) as residues that neighboured active residues having > 20% side chain accessible surface area and had chemical shift changes greater than average. This approach led to a set of ambiguous restraints between UbcH7 (K9, A59, E60, F63, K64, E93, N94, K96, A98) and R0RBR parkin (T242, L266, T270, Q276, A291, G292, Q389-D394, R396) and between Ub (F4, T7-G10, I11, E34, I36, V70, L71, L73, G75, G76) and R0RBR parkin (Q276, V330, L331, R366, A379-S384,G385, T386).

The UbcH7-Ub conjugate was docked to R0RBR:pUb using HADDOCK (Dominguez *et al*, 2003) using the residues described above. Starting coordinates from the crystal structure of pUb: UblR0RBR (PDB code 5N2W) were used following removal of the Ubl domain and adjoining linker coordinates (residues 1–83). Several linker sections absent in crystal structures of parkin were incorporated using the Modeller (Eswar *et al*, 2006) plug-in for UCSF Chimera (Pettersen *et al*, 2004). The tether (387–405) that partly occludes the RING1 binding site in the starting coordinates was allowed to move. Coordinates for ubiquitin (PDB code 1UBQ;

Vijay-Kumar *et al*, 1987) and UbcH7 (PDB code 4Q5E; Grishin *et al*, 2014) were used to sequentially dock the UbcH7 and Ub moieties untethered according to restraints and using a single unambiguous restraint between the C-terminal G76 of ubiquitin and the catalytic C86K of UbcH7 to create the UbcH7-Ub conjugate in the complex. Upper distance limits of 4.0 Å were set for ambiguous distance restraints, while the unambiguous distance restraint was set to 6.8 Å. Standard HADDOCK parameters were used except inter_rigid (0.1) which was set to allow tighter packing of the proteins, and the unambiguous force constant was set fivefold higher compared to those used the ambiguous constants. A total of 1,000 initial complexes were calculated, and the best 100 structures were water-refined.

## Data availability

Coordinates for models of the UbcH7-Ub conjugate bound to R0RBR:pUb reported here have been deposited to the Protein Data Bank under accession number 6N13.

**Expanded View** for this article is available online.

## Acknowledgements

This work was supported by postgraduate scholarships from the Natural Sciences and Engineering Research Council of Canada (TECC, KMD), a collaborative travel grant from Boehringer-Ingelheim Fonds (TECC), research grants from the Canadian Institutes of Health Research (MOP-14606) to GSS, Medical Research Council (MC_UU_12016/12) to HW and Natural Sciences and Engineering Research Council of Canada (RGPIN-2018-04243) to LK, and a Wellcome Trust Investigator Award (209347/Z/17/Z) to HW. The authors would like to thank the reviewers for their many helpful suggestions, Dr. Martin Duennwald for critical reading of this manuscript and Dr. Liliana Santamaria for maintenance of the Biomolecular NMR Facility at Western.

## Author contributions

TECC collected and assigned all NMR experiments, completed and analysed HADDOCK modelling for the E2-Ub complex with pParkin:pUb, and wrote the manuscript. KMD conducted ubiquitination assays and NMR-based interaction experiments, completed UbVS reactivity experiments, and analysed and wrote the manuscript. EAF did HDX experiments and NMR interaction experiments, analysed the data and wrote the manuscript. KRB expressed and purified proteins for ubiquitination assays, NMR experiments and HDX experiments and wrote the manuscript. JDA helped with NMR-based interaction experiments. VKC completed ubiquitin loading and off-loading experiments and wrote the manuscript. YX and LK helped with the design and analysis of HDX experiments. HW and GSS conceived the study, designed experiments, analysed the data and wrote the manuscript.

## Conflict of interest

The authors declare that they have no conflict of interest.

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
