## [Review Process File · The EMBO Journal]

Synergistic Recruitment of UbcH7~Ub and Phosphorylated Ubl Domain Triggers Parkin Activation

Tara E.C. Condos, Karen M. Dunkerley, E. Aisha Freeman, Kathryn R. Barber, Jacob D. Aguirre, Viduth K. Chaugule, Yiming Xiao, Lars Konermann, Helen Walden and Gary S. Shaw.

Review timeline:

Submission date:	8 th June 2018
Editorial Decision:	10 th July 2018
Revision received:	28 th September 2018
Editorial Decision:	24 th October 2018
Revision received:	25 th October 2018
Accepted:	26 th October 2018

Editor: Hartmut Vordermaier.

Transaction Report:

1st Editorial Decision

10th July 2018

Thank you again for submitting your manuscript on UbcH7 recruitment and Parkin activation to The EMBO Journal. It has now been seen by three expert referees, and given their overall supportive opinions, I would like to invite you to prepare a revised manuscript, addressing the various comments from all three reviewers. Among other issues, you will see that one of the main concerns is that the data obtained should be fleshed out more deeply, also in comparison with other recently published work (Gladkova et al, Sauve et al), in order to derive most conclusive insights.

Should you be able to satisfactorily address the various experimental and presentational issues noted by all three referees, we should be able to consider a revised manuscript further for publication in The EMBO Journal.

REFERE REPORTS

Referee #1:

Parkin is a member of the RBR family of E3 ubiquitin ligases and plays a major role during mitophagy by promoting the turnover of mitochondrial proteins. Misregulation and/or dysfunction of Parkin activity is implicated in autosomal recessive juvenile Parkinsonism (AR-JP), with disease variants throughout Parkin's multiple domains, including a RING1 domain that interacts with E2~Ub, and a catalytic RING2 domain that forms an obligate E3~Ub intermediate. Like other RBRs, Parkin is auto-inhibited by intra-domain architecture that must be reorganized following distinct activation events prior to catalysis. The authors build on previous findings in which activation of Parkin requires a PINK1 kinase-mediated phosphorylation of the N-terminal Ubl domain and binding of phospho-Ub, by providing evidence for additional domain remodeling following E2~Ub binding that leads to an optimal activation state. NMR experiments reveal that Parkin recruits the E2~Ub conjugate in an 'open' conformation, a consistent theme seen for other

RBRs, HHARI and HOIP. Domain remodeling is assessed via NMR chemical shift analysis and hydrogen deuterium exchange mass spectrometry (HDX/MS), identifying E2~Ub induced conformational changes in Parkin that promote phospho-Ubl recruitment to the R0 domain, enhancing catalysis.

With regards to the very recent Komander publication, which uses HDX and crystallography to capture Parkin in an 'activated' state, the study reported here adds both information and insight. However, its present form falls short of delivering its full potential to define the critical dynamic properties and events that underlie the activation of Parkin ligase activity. Importantly, the NMR and HDX/MS approaches can report directly on dynamics whereas the crystallographic studies can only imply/infer dynamic events. The present manuscript will be strengthened and be of higher impact if the authors take the opportunity to fully compare their observations with those of the Komander team. Both studies suggest remodeling of the R0:RING2 interaction occurs upon Parkin activation, but they seem to differ on both the timing and extent of activating changes. Komander used HDX data and previous failed attempts to crystalize active Parkin to propose a model in which RING2 becomes completely exposed in the pParkin:pUb state for subsequent interaction with E2-Ub. While C-terminal peptides (r450+) of RING2 are more exposed in the presence of E2~Ub & UbVS in the Komander HDX data, a majority of RING2 & REP-RING2 linker peptides were 'not able to be analyzed' or show no change in exchange in the pUbl:pUb state. Additionally, Komander ExtDataFig2(5) shows peptides for the RING2:R0 hydrophobic interaction, with (by eye) RING2 c-term exposure increasing by:

- ♣ Parkin vs. pParkin:pUb ~2-fold
- ♣ pParkin:pUb vs. pParkin:pUb:E2-Ub ~2-fold
- ♣ pParkin:pUb vs. pParkin-UbVS:pUb ~4 fold

The additional increase induced by E2 binding seems consistent with the conclusions presented by Shaw and co-workers, in which additional remodeling of the R0/R2 site occurs upon E2~Ub binding, but the effect is somewhat glossed over in the Komander paper. Neither study fully addresses what is happening to RING2, but this important step could be clarified using techniques presented in this work. For example, a comparison of HSQC experiments of Parkin:pUb to pParkin:pUb to the TEV-cleaved-RING2 pParkin:pUb (DK) would provide insight into whether the RING2 domain is tumbling more independently in the pParkin:pUb state or remains weakly bound, and additionally whether the E2 promotes this RING2 dissociation. Presumably the TEV linker is sufficiently accessible to be cleaved, but the orientation of RING2 to the rest of Parkin remains unclear. Additionally, NOE, T1, T2 experiments of full-length Parkin may further define the flexibility of the REP-RING2 linker.

While the data and final model are mostly convincing, some experiments leave room for other interpretations and could be complemented by additional data. Notably, the authors present an impressive toolbox of techniques, but unfortunately, they do not apply to all steps of their model (e.g. HDX/MS is presented only for auto-inhibited and pParkin/pUb, but not for pParkin/pUb+UbC^{H7}-Ub).

Overall, the authors present a comprehensible model for the mechanism of activation in the final figure, but the order and logic of figures leading up to the mechanism do not systematically address each step of the model. A more logical order of presentation of data figures that more clearly supports and builds up to the model would improve the readability, and therefore, impact, of the manuscript. Finally, the manuscript is written in a terse manner that assumes that a reader is well versed in both the details of Parkin-ology and in the analysis and interpretation of NMR and HDX/MS data. In sum, the manuscript makes a worthy contribution to our understanding of how Parkin is activated, but inclusion of additional data (see below), reorganization of the text and figures, and editing for better readability will improve the paper.

Specific suggestions (outlined by Figures):

Figure 1. Model of the E2-Ub Conjugate bound to pUb-activated parkin.

- Non-expert readers would benefit from a block diagram of Parkin to introduce nomenclature and domain connections.
- Potential reorganization of figures: consider presenting the HDX data first to describe full-length Parkin conformational changes under previously defined activation events (pUb binding & pUbl).

This will set the stage for E2~Ub-mediated changes described here. In that sense, the content of Fig.1 could become Fig.2

- NMR data in Supplemental Figure 1A is important to include as a panel in Figure 1.
- Labeling in Figure 1C is excessive and confusing. Highlight a select few and maintain the domain color scheme in the text.
- The importance and relevance of the chosen AR-JP mutants in Figure E is unclear. It would strengthen the authors' message and would be highly interesting to the field if the authors could identify one (or more) AR-JP mutant that has a defect in the particular activation mechanism described in this paper. If possible, the authors should choose one such mutant and analyze it fully.
- The peptide array is confirmatory and can be reformatted as a supplemental figure.
- More explanation of how the E2~Ub : Parkin Haddock model constraints were generated is necessary, given that the placement of the ubiquitin conjugate compared to other RBR:E2~Ub orientations is presented and discussed (Supplemental Figure 4). Experiments involving unconjugated UbcH7 and free Ub would address the orientation and interaction sites of such a proposed model. It is unclear, for example, what constraints lead to the position and contacts of the conjugated Ub in the model. How were CSPs observed in E2~Ub upon binding to parkin parsed in terms of which are due to the opening of the E2~Ub and which are due to contacts made with Parkin?

Figure 2. Remodelling of the RING0/RING2(Rcat) Interface induced by UbcH7-Ub Interaction.

- Additional labeling in Figure 2B is needed to orient the reader - highlight W403.

Figure 3. Ubl Phosphorylation and E2-Ub binding alter exposure of parkin's catalytic cysteine.

- The logic of using Ub-VS as a probe for active site accessibility is unclear. Why was a smaller chemical probe not used? The bulkiness of a Ub-VS adduct for active site accessibility assays is concerning for steric reasons, given the close proximity to proposed Ubl binding sites. In this context, the authors should emphasize that Ub-VS does not mimic a the catalytically relevant Ub (which is the one attached to UbcH7), but rather a "bulky" chemical probe for accessibility of the RING2 cat. cysteine.
- The representation of panel B/D is misleading given the differences in time scale for full-length and Δ Ubl assays.
- Assessing Ub-VS reactivity with a W403A mutant could address whether the R0/R2 interface is altered and thus promotes active site accessibility.
- A uniform and comprehensive nomenclature for a thioester and lysine-linked Ub should be adopted throughout the paper (e.g. UbcH7~Ub = thioester, UbcH7-Ub = lysine-linked)

Figure 4. Weak interaction of Ubl with parkin after phosphorylation.

- Reorganize panels so that the pUbl in the context of full-length Parkin is presented first.
- There appears to be peak doubling in the spectrum of full-length Parkin (Q25, Q63). If true, this is intriguing and (possibly) consistent with the model of "partial activation" presented by the authors. Is there additional evidence from the full spectrum to suggest that the apparent peak doubling could be due to multiple Ubl states, while the addition of E2~Ub promotes a subset of those sampled states (supporting the model shown in Figure 6)?
- A complementary and potentially powerful experiment would be to segmentally label the Ubl in the context of Parkin to unambiguously observe its resonances under a variety of relevant conditions/states. A similar strategy to observe RING2 resonances would also be fascinating, though potentially outside the scope of this study.

Figure 5. Identification of a pUbl binding site and exposure of RING2(Rcat) C-terminus.

- Additional HDX experiments including E2~Ub would help support the authors model of domain rearrangements, however, in light of the recent Komander paper, which presented HDX on 4 Parkin species: pUb binding, pUbl, E2~Ub binding, Cat-UbVS, this may not provide sufficient novel information to warrant the substantial work required.
- Presenting the HDX data earlier in the paper would be a better use of the current dataset given the current lack of E2~Ub HDX data.

- Presentation of the HDX data should be expanded: the profiles for each individual state that is compared in the currently presented data should be shown, and experimental details such as peptide coverage must be presented so that readers can more critically assess the data and its interpretation.
- Figure 5B heat map coloring needs to be explained.

Figure 6. Model of parkin activation by combined pUbl and E2~Ub interactions.

- Contrasting colors should be used to denote phosphorylation and the catalytic cysteine; the current coloring scheme is confusing.

Referee #2:

Condos et al. report a very thorough investigation of the steps involved in activation of Parkin by phosphorylation of the Ubl domain within Parkin and the binding of UbcH7~Ub. This study utilizes a variety of structural and biophysical methods to gain insight into the different interactions between the core of Parkin (R0RBR) with Ubl, pUbl, Ub, pUb and UbcH7~Ub as the complex is progressively assembled and activated. The experiments reflect solution state complexes using NMR and drawing upon the assembled knowledge of Parkin, its domains, and complexes that have been reported in numerous crystallographic studies. The combination of NMR with functional assays and hydrogen-exchange mass spectrometry lead to a proposed model of the dynamic interactions and remodeling associated in transitioning from the auto-inhibited state to the fully activated state. The model suggests that some of the interfaces that are active and modulated in this process correspond to known mutations in ARJP disease.

The model provides a hypothesis for the role of PINK1 phosphorylation of the Ubl domain and the role of phosphorylated Ub in achieving the fully active E2~Ub bound state of Parkin. The detailed residue-level examination of interactions via NMR methods enables a stepwise examination of the interfaces between the relative domains and rearrangements required to support these interactions. Correlation of the findings with functional assays and comparison to other data for mutations is strong. It is a well-constructed model built on clever deconstruction of the interactions. While the study encompasses some divide-and-conquer strategies, the authors are careful to use the most complete components of the systems as is feasible and which is relevant to the interactions under study.

The conclusions and the model are well supported by the data and provide new insights in to the activation process of Parkin. While there is more to be elucidated, this study lays out the outline of the process and will support further research by both these groups and others.

Comments:

Page 7: Middle paragraph: suggest that the sentence should read "...RORBR shift in the presence of E2-Ub and ..."

Page 7, bottom paragraph: The input data for the HADDOCK calculation should be given some scale in the text, which is in addition to the methods. The number of restraints or contacts observed and used in the calculation should be provided. This would be a summary of the total in the text and in the respective categories in the Methods section (page 18).

Page 8, Figure 1D and Methods: The experiment reported in Figure 1D is insufficiently described. Is this a protection assay for binding of UbcH7-Ub to RING1 and IBR?

Page 9 bottom paragraph: The observation of residue W403 as an anchor for the REP element should be confirmed by a significant effect on the indole sidechain resonance of residue W403. Is this observed? If not, why? Also, residue V465 is explicitly mentioned, but it does not appear to be visible in Panel B.

Page 9-10: Figure 2. The peaks indicated in Panel A correspond to residues that should be visible (at least some of them) in Panel B. The figure should be adjusted to highlight these residues in Panel B.

Page 10: The postulation for losing signals in the RORBR:pUb + UbcH7-Ub complex is that a remodeling occurs. Is this due to chemical exchange effects that result from indirect or allosteric effects? Is this seen at all molar ratios of the complex, or is it possible to observe the shift at a low molar ratio followed by disappearance at a higher molar ratio? Could the exchange effect also be confirmed by either a temperature or field dependence effect?

Page 11 bottom paragraph: Figure 4A/B the ratio of components is stated as equimolar for panel A but not stated for panel B. This should be specified to support and enable interpretation of the 'fast-exchange' condition stated in the text on page 11-12.

Page 12 and Figure 5B: The color coding in Figure 5B is not described. Presumably, it corresponds to the peptides with large HDX factors that are described in the text. Some color code needs to be provided in either the text or the figure legend.

Page 13: ...is blocked; however, static.... Use semicolon

Page 13: The PDB codes for some of the structures should be listed in the figure legends or the text somewhere. The first paragraph of the Discussion refers to them, and there are multiple structures used to make the figures, so citation of the PDBID should exist somewhere in the manuscript. This is done in part in the methods, but the entire manuscript should be reviewed for consistency on this point.

Page 16-18: There are numerous reagents for which a source or description is not available. For example, UbVS, Ub800, etc.

Page 26: the legend to Figure 4 does not indicate the changes in the rest of the spectrum. Were these the only residues that shifted or were there some apparently non-specific effects? In panel B, the experiment is a mixture of ¹⁵N-labelled pParkin and pUb, but the molar ratio is not defined. This ratio should be included. Do the shifted peaks completely disappear at a higher ratio and/or are more peaks observed at a smaller ratio?

Page 27: Legend to Figure 5 B. The various colored segments in the structure are not described, nor are they described in the text. Since these colors are intended to reveal aspects of the interaction, then they should be described in either the main text or in the legend.

Supp. Fig 5: the domain names RING0, RING1, etc. are rather weak and would benefit from a bold font.

Supp. Fig 7: add an x-axis label of (min)

Referee #3:

In this manuscript, Condos et al, study the mechanism of UbcH7-Ub recruitment to the RBR E3 parkin in the context of phosphorylation events in Ubl and Ub that relieve E3 autoinhibition. The authors used chemical shift perturbation data to create a model of a UbcH7-Ub bound to RORBR:pUb and present additional data suggesting that parkin recruits UbcH7-Ub in an open conformation, that E2-Ub binding to parkin remodels the RING0/RING2 interface and enhances the reactivity of parkin's catalytic cysteine. Overall, this study provides some novel insights into Parkin mechanism, however, there are several important issues that need to be addressed in order to clarify and strengthen the author's major conclusions and elevate the manuscript to a state more suitable for publication in EMBO. Lastly, Gladkova et al., recently published a manuscript describing the structure of phosphorylated parkin that has several overlapping elements with this study. The data and conclusions from the Gladkova et al. study should be considered and discussed.

Comments:

- The recent Gladkova et al. paper describing the structure of phosphorylated parkin should be referenced. How the HADDOCK model and other data presented in this paper compare to PDB:6GLC and the HDX data presented in the Gladkova et al. study should be discussed.
- Where were the backbone resonance assignments of the R0RBR Parkin fragment obtained? Surely these assignments were not trivial and this data underlies many of the CSP experiments, including the underlying data used to generate the HADDOCK model. However, there is no detail of how these assignments were performed/validated in the Methods, there is no citation in which these details were provided, nor is there an R0RBR entry in the Biological Magnetic Resonance Data Bank. Will the assignments and CSPs be deposited in the BMRB?
- Figure 3A,C. The author's model for why UbcH7-Ub binding decreases reactivity of parkin's catalytic cysteine in the context of full length pParkin/pUb/E2-Ub is hard to understand. Wouldn't this data suggest that pUbl interactions with RING0 attenuate parkin activity in the presence of UbcH7-Ub rather than 'act synergistically to drive parkin activity' as the authors state? Further, the total levels of UbVS modification occurring in full length pParkin/pUb/UbcH7-Ub and R0RBR/pUb/pUbl/UbcH7-Ub are quite similar. If pUbl and UbcH7-Ub binding were acting synergistically to drive parkin activity wouldn't the total levels of UbV modification be expected to be greater in the context of full length pParkin? Might a kinetic analysis might be more informative?
- Were the experiments in Figure 3 performed in triplicate? Were C431A negative controls performed to ensure the crosslinks are specific to the catalytic cysteine, particularly since there are several Ub binding sites? Were these experiments performed with the addition of free UbcH7? This latter experiment could provide some interesting details into the mechanism of the author's observations. Are the same results obtained in the presence of UbcH5-Ub?
- Supplemental Figure 1 displays a line that indicates the average CSP +/- 1 standard deviation, however, many of the residues discussed in the manuscript fall well below this threshold. What are the criteria by which the authors determined 'significance'?
- Along these lines, the region between Parkin residues 230 and 240 harbors one of the largest clusterings of residues that experience CSPs above 1 standard deviation upon UbcH7-Ub binding. What accounts for the large chemical shifts in this region? Do these residues contact UbcH7 or Ub in the HADDOCK model? Were they included as 'interacting residues' during the HADDOCK calculations?
- The most novel aspect of the HADDOCK model is the parkin binding site for Ub from the UbcH7~Ub thioester mimetic. While the author's tested ARJP and non-ARJP mutations on the parkin side and found a loss of parkin activity, no Ub mutants at the predicted parkin/Ub interface were presented to validate the position of Ub in the model through either loss of function or gain of function in the context of the above parkin mutants. In the absence of biochemical data in support of the proposed Ub binding site, is it possible to perform the CSP experiments with free UbcH7 to determine whether the CSPs resulting from contacts between Ub and the putative Ub binding site on parkin are present? In addition to providing evidence in support of the author's HADDOCK model, this experiment would potentially illuminate the role of E2 binding versus Ub binding in triggering remodeling of the RING0/RING2 interface
- Martino et. al recently demonstrated that UbcH7-Ub engages the RBR HHARI/ARIH1 differently from UbcH5-Ub. Have the author's performed the CSP experiments with UbcH5-Ub?
- Page 14, third sentence from the bottom- issue with logic? The author's posit that rapid association/dissociation of pUbl with R0RBR:pUb is to ensure parkin activity only in the presence of the loaded E2~Ub, but parkin can only be active upon charging with Ub after E2~Ub has already bound.
- The last sentence of page 8 leading into page 9 is quite difficult to follow and the accompanying figure does not clearly show what is being stated in the text. Also, what putative interactions lock the flexible Ub C-terminus in an orientation in which G76 carboxyl group is 'exposed' towards the tether side of parkin? There do not appear to be any parkin residues in proximity.

- The abstract is misleading. 'We present the structure of an incoming E2~Ub conjugate with phospho-ubiquitin bound parkin R0RBR' should be clarified. The structure that is presented is a HADDOCK model rather than an experimentally derived structure. The title of Figure 1 or the sentence on page 7 stating "We used chemical shift perturbation results to determine a model of a UbH7-Ub bound to R0RBR:pUb using HADDOCK" is more accurate.
- Typo page 8, line 11? Is V293 correct?

1st Revision - authors' response

28th September 2018

Point-by-point response

Editorial Comments

Among other issues, you will see that one of the main concerns is that the data obtained should be fleshed out more deeply, also in comparison with other recently published work (Gladkova et al, Sauve et al), in order to derive most conclusive insights.

We have re-written the discussion to include a comparison to these recent crystal structures. In many ways these structures have helped us strengthen our model.

Response to Reviewers

Reviewer 1

With regards to the very recent Komander publication, which uses HDX and crystallography to capture Parkin in an 'activated' state, the study reported here adds both information and insight. However, its present form falls short of delivering its full potential to define the critical dynamic properties and events that underlie the activation of Parkin ligase activity. Importantly, the NMR and HDX/MS approaches can report directly on dynamics whereas the crystallographic studies can only imply/infer dynamic events. The present manuscript will be strengthened and be of higher impact if the authors take the opportunity to fully compare their observations with those of the Komander team. Both studies suggest remodeling of the R0:RING2 interaction occurs upon Parkin activation, but they seem to differ on both the timing and extent of activating changes. Komander used HDX data and previous failed attempts to crystalize active Parkin to propose a model in which RING2 becomes completely exposed in the pParkin:pUb state for subsequent interaction with E2-Ub. While C-terminal peptides (r450+) of RING2 are more exposed in the presence of E2~Ub & UbVS in the Komander HDX data, a majority of RING2 & REP-RING2 linker peptides were 'not able to be analyzed' or show no change in exchange in the pUbl:pUb state.

We have used the Discussion to compare our HDX data with that from both the Sauvé and Gladkova papers. In order to do this we completed further HDX experiments using pParkin:pUb bound to non-hydrolyzable UbH7-Ub. This latter data is now presented in Fig 4 while our original HDX data using pParkin:pUb now comprises Fig. 1. Unlike both the other studies we were able to obtain >91% peptide coverage of parkin in all of our HDX experiments and can therefore report on regions lacking in the other studies. We would like to point out that some differences exist in the way each group analyzed (ie. basis data set, time point considered) and presented their data (bar graph, heat map, mapped on structure). In our experiments we plotted %²H uptake difference between parkin and pParkin:pUb (Fig 1C) and %²H uptake difference between pParkin:pUb and pParkin:pUb:UbH7-Ub (Fig 4A) in a similar manner as the Komander group. A significant difference between our data is that we were able to measure HDX for residues 60-80 (Ubl, including pSer site) and 410-440 (RING2(Rcat)) in our experiments. As with the Komander group we see exposure of the extreme C-terminus in pParkin:pUb although the remainder of RING2(Rcat) is still buried. Upon E2-Ub addition, we see complete exposure of the entire RING2(Rcat) (no data in Komander paper, not done in Sauvé paper) and a small protection of its C-terminus. We attribute this to the release of RING2(Rcat) in the presence of both phosphorylation and E2-Ub binding and

hypothesize that the RING2(Rcat) becomes protected due to interaction with the conjugated Ub, analogous to the HOIP:UbcH5b-Ub structure.

Additionally, Komander ExtDataFig2(5) shows peptides for the RING2:R0 hydrophobic interaction, with (by eye) RING2 c-term exposure increasing by:

*Parkin vs. pParkin:pUb ~2-fold
pParkin:pUb vs. pParkin:pUb:E2-Ub ~2-fold
pParkin:pUb vs. pParkin-UbVS:pUb ~4 fold*

The additional increase induced by E2 binding seems consistent with the conclusions presented by Shaw and co-workers, in which additional remodeling of the R0/R2 site occurs upon E2~Ub binding, but the effect is somewhat glossed over in the Komander paper.

For consistency we completed our own HDX experiments to examine these differences. We find that this short C-terminal region becomes exposed in the pParkin:pUb case in line with the Gladkova work and the Sauvé work although our peptides are all not quite the same. When we did experiments with E2-Ub we observed a small protection of this region but complete exposure of the RING2(Rcat) not analyzed by Gladkova. So this is consistent with re-modelling of the RING0/RING2(Rcat) region.

Neither study fully addresses what is happening to RING2, but this important step could be clarified using techniques presented in this work. For example, a comparison of HSQC experiments of Parkin:pUb to pParkin:pUb to the TEV-cleaved-RING2 pParkin:pUb (DK) would provide insight into whether the RING2 domain is tumbling more independently in the pParkin:pUb state or remains weakly bound, and additionally whether the E2 promotes this RING2 dissociation. Presumably the TEV linker is sufficiently accessible to be cleaved, but the orientation of RING2 to the rest of Parkin remains unclear. Additionally, NOE, T1, T2 experiments of full-length Parkin may further define the flexibility of the REP-RING2 linker.

This is an excellent suggestion and we agree that identifying the orientation of the RING2(Rcat) domain is important but not yet clear. As mentioned above we feel our HDX experiments were able to measure segments not analyzed by the other groups provides very good evidence that the RING2(Rcat) is liberated only in the presence of the E2-Ub and phosphorylation. In principle, the suggested NMR relaxation experiments using Parkin:pUb, pParkin:pUb and TEV-cleaved-RING2 pParkin:pUb presumably +/- E2-Ub might be able to provide evidence for dissociation of the RING2(Rcat) domain. In practice these experiments using TEV-cleaved-RING2 pParkin would require NMR re-assignment of much of the parkin protein (since the RING2 and tether has been removed). The size of the complex, low concentrations we work with (typically 100 uM), and the instability of the complexes in the NMR tubes have also presented problems. It is clear that the truncated form of parkin is rather poorly behaved (even though it was crystallized) based on comments in both the Sauvé and Komander papers. Both are impressive achievements.

In place of this we would like to draw attention to ¹H-¹⁵N heteronuclear nOe experiments on R0RBR:pUb (Fig EV1) that we have been able to acquire for this paper and R0RBR parkin (previously published, EMBO J (2015)). We have tried to highlight these dynamics experiments better in the text. Unlike the available crystallographic data, these data show that several sections of parkin are flexible. For example, the entire IBR domain is flexible in the absence of pUb binding. Large sections of the tether region are more flexible in the absence of pUb binding and curiously a small region near the phospho-site in RING0 domain (around residues 155-160) changes flexibility upon pUb binding.

While the data and final model are mostly convincing, some experiments leave room for other interpretations and could be complemented by additional data. Notably, the authors present an impressive toolbox of techniques, but unfortunately, they do not apply to all steps of their model (e.g. HDX/MS is presented only for auto-inhibited and pParkin/pUb, but not for pParkin/pUb+UbcH7-Ub).

We have now completed HDX experiments for pParkin/pUb+UbcH7-Ub (Fig. 4A, B).

Overall, the authors present a comprehensible model for the mechanism of activation in the final figure, but the order and logic of figures leading up to the mechanism do not systematically address each step of the model. A more logical order of presentation of data figures that more clearly supports and builds up to the model would improve the readability, and therefore, impact, of the manuscript. Finally, the manuscript is written in a terse manner that assumes that a reader is well versed in both the details of Parkin-ology and in the analysis and interpretation of NMR and HDX/MS data. In sum, the manuscript makes a worthy contribution to our understanding of how Parkin is activated, but inclusion of additional data (see below), reorganization of the text and figures, and editing for better readability will improve the paper.

We would like to thank the reviewer for this helpful and insightful suggestion. As a result we have completely reorganized the manuscript similar to that suggested below. We have added new schematics to several figures to help readers less familiar to the parkin structure and have included new details of the HDX/MS and NMR data.

Specific suggestions (outlined by Figures):

Figure 1. Model of the E2-Ub Conjugate bound to pUb-activated parkin.

- *Non-expert readers would benefit from a block diagram of Parkin to introduce nomenclature and domain connections.*
- *Potential reorganization of figures: consider presenting the HDX data first to describe full-length Parkin conformational changes under previously defined activation events (pUb binding & pUbl). This will set the stage for E2~Ub-mediated changes described here. In that sense, the content of Fig.1 could become Fig.2*

We have reorganized Fig. 1 as suggested and started with a description of auto-inhibited parkin and pUb-bound (optimized) parkin (Fig 1A). We have included small block diagrams here and in other figures to show the domain organization of parkin. The HDX experiments for pParkin:pUb vs parkin are now in Fig B-D.

- *NMR data in Supplemental Figure 1A is important to include as a panel in Figure 1.*
- *Labeling in Figure 1C is excessive and confusing. Highlight a select few and maintain the domain color scheme in the text.*

UbcH7-Ub bound to R0RBR:pUb is now placed in Fig. 2. We have moved the NMR data from Supplemental Fig 1A to Fig 2A, Upon reflection we felt the previous Fig 1C was excessive and the information could be garnered from our new Fig 2 and Fig 3B, so this figure was removed.

- *The importance and relevance of the chosen AR-JP mutants in Figure E is unclear. It would strengthen the authors' message and would be highly interesting to the field if the authors could identify one (or more) AR-JP mutant that has a defect in the particular activation mechanism described in this paper. If possible, the authors should choose one such mutant and analyze it fully.*

We chose substitutions comprising a few AR-JP mutants and others to test our E2 and Ub binding sites. We have tried to clarify this in the text. We have conducted new experiments using substitutions in Ub to test the conjugated Ub binding site from our model. This data is now presented in Fig 2D.

- *The peptide array is confirmatory and can be reformatted as a supplemental figure.*

We agree that the peptide array did not offer new information and have removed this data from the paper.

- *More explanation of how the E2~Ub : Parkin Haddock model constraints were generated is necessary, given that the placement of the ubiquitin conjugate compared to other RBR:E2~Ub orientations is presented and discussed (Supplemental Figure 4). Experiments involving unconjugated UbcH7 and free Ub would address the orientation and interaction sites of such a proposed model. It is unclear, for example, what constraints lead to the position and contacts of the conjugated Ub in the model. How were CSPs observed in E2~Ub upon binding to parkin parsed in*

terms of which are due to the opening of the E2~Ub and which are due to contacts made with Parkin?

We have included more information on experiments that were used to measure CSPs and generate HADDOCK restraints in the text, figure legends and Methods sections. In general HADDOCK restraints were generated according to the standard HADDOCK protocol. However in our case, identification of a “significant” CSP was based on its absence in the bound state, indicating it underwent a large change in position (ie. grey bars in Fig EV2, Appendix Fig S3) or the signal shifted more than one standard deviation above the average shift (dashed lines in same figures). These residues were filtered for accessible surface area according to the HADDOCK protocol and used as unambiguous restraints. We collected an individual experiment that monitored UbcH7 binding only to R0RBR:pUb (Fig EV2B). Though not as impressive, likely due to a weaker binding, this data clearly showed regions that aligned with similar portions as in UbcH7-Ub. We were unable to complete a similar experiment with Ub due to its very weak affinity (~40-60 uM) and the low concentrations of R0RBR:pUb we were working with. Residues for the closed vs open forms of E2 and Ub were subjected to the same selection process. Most of these CSPs were very small (previously noted by Dove et al (2016)), fell outside the threshold (average + 1SD) for binding and did not experience line broadening.

Figure 2. Remodelling of the RING0/RING2(Rcat) Interface induced by UbcH7-Ub Interaction.

- Additional labeling in Figure 2B is needed to orient the reader - highlight W403.

This is now Fig 3B. We have re-coloured the figure and highlighted W403 in red.

Figure 3. Ubl Phosphorylation and E2-Ub binding alter exposure of parkin's catalytic cysteine.

- The logic of using Ub-VS as a probe for active site accessibility is unclear. Why was a smaller chemical probe not used? The bulkiness of a Ub-VS adduct for active site accessibility assays is concerning for steric reasons, given the close proximity to proposed Ubl binding sites. In this context, the authors should emphasize that Ub-VS does not mimic a the catalytically relevant Ub (which is the one attached to UbcH7), but rather a "bulky" chemical probe for accessibility of the RING2 cat. cysteine.
- The representation of panel B/D is misleading given the differences in time scale for full-length and Δ Ubl assays.
- Assessing Ub-VS reactivity with a W403A mutant could address whether the R0/R2 interface is altered and thus promotes active site accessibility.
- A uniform and comprehensive nomenclature for a thioester and lysine-linked Ub should be adopted throughout the paper (e.g. UbcH7~Ub = thioester, UbcH7-Ub = lysine-linked)

This data is now presented in Fig. 5 and Appendix Fig. S7. We added new text (p16) to rationalize the selection of Ub-VS. We used this probe because it has been used many times to test parkin reactivity including the recent Gladkova paper, so we tried to stay consistent. We have highlighted newer probes (Pao et al, 2016) in the manuscript. We, and others, have shown the Ub-VS is selective for C431 in human parkin. We have corrected the assays in Fig 5 so that all experiments now reflect the same time scale. We have completed assays using the W403A substituted parkin (Appendix Fig 7C) and show this substitution does show low basal reactivity. Finally we have corrected and tried to be more consistent with the use of UbcH7~Ub vs UbcH7-Ub.

Figure 4. Weak interaction of Ubl with parkin after phosphorylation.

- Reorganize panels so that the pUbl in the context of full-length Parkin is presented first.
- There appears to be peak doubling in the spectrum of full-length Parkin (Q25, Q63). If true, this is intriguing and (possibly) consistent with the model of "partial activation" presented by the authors. Is there additional evidence from the full spectrum to suggest that the apparent peak doubling could be due to multiple Ubl states, while the addition of E2~Ub promotes a subset of those sampled states (supporting the model shown in Figure 6)?
- A complementary and potentially powerful experiment would be to segmentally label the Ubl in the context of Parkin to unambiguously observe its resonances under a variety of relevant conditions/states. A similar strategy to observe RING2 resonances would also be fascinating, though potentially outside the scope of this study.

This data has now been re-arranged as suggested (full-length parkin first) and is now Fig. 4C,D. There is no signal doubling in the spectrum although it may appear that way. The spectrum of full-length pParkin:pUb is more complicated than pUbl alone because it contains many more signals, most of which are from flexible portions of the protein. A modest number of signals shift near pSer65 in pParkin compared to pUbl alone while most other signals do not shift. It is important to note that the ^1H - ^{15}N HSQC spectrum shown here for full-length parkin (465 residues) was collected using a 30 msec T2 filter. This relaxation method allows only resonances to be visualized from more flexible portions of the protein. We have previously published the details of this work (PNAS, 2017). Using this experiment we are able to visualize and assign the entire pUbl domain within pParkin (no segmental labelling required) while the remainder of the signals in parkin are essentially invisible. This experiment indicates that the pUbl domain is not fully bound to the remainder of parkin (or its signals would not be visible). The experiment with the RING2(Rcat) is a good suggestion, akin to the Komander TEV-cleaved-RING2 pParkin experiments mentioned above and we agree it might be useful in a future study.

Figure 5. Identification of a pUbl binding site and exposure of RING2(Rcat) C-terminus.

- *Additional HDX experiments including E2~Ub would help support the authors model of domain rearrangements, however, in light of the recent Komander paper, which presented HDX on 4 Parkin species: pUb binding, pUbl, E2~Ub binding, Cat-UbVS, this may not provide sufficient novel information to warrant the substantial work required.*
- *Presenting the HDX data earlier in the paper would be a better use of the current dataset given the current lack of E2~Ub HDX data.*
 - *Presentation of the HDX data should be expanded: the profiles for each individual state that is compared in the currently presented data should be shown, and experimental details such as peptide coverage must be presented so that readers can more critically assess the data and its interpretation.*
- *Figure 5B heat map coloring needs to be explained.*

The HDX experiments referred to above, for pParkin:pUb compared to autoinhibited parkin, have been moved to Fig. 1. We have added new HDX experiments that compare pParkin:pUb:UbcH7-Ub with pParkin:pUb in order to identify how the E2-Ub might alter the conformation of parkin. Although a great deal of work, we felt it was important to include our results because the Gladkova paper was unable to map some of the most informative regions in the Ubl and RING2(Rcat) domains. Our data shows that most of the RING2(Rcat) domain is not exposed in the pParkin:pUb state but exposed in upon E2-Ub binding. We also felt it was important for us to complete these experiments to produce a more well-rounded manuscript as indicated from an earlier comment. Further it is clear that some differences exist between our data and the Gladkova and Sauvé data due to slightly different methods of analysis. We have included new figures to show peptide coverage (Appendix Fig S2) and selected ^2H uptake curves (Fig. 1B, Appendix Fig S1, S6). The heat maps in HDX figures have been explained in the figure legends.

Figure 6. Model of parkin activation by combined pUbl and E2~Ub interactions.

- *Contrasting colors should be used to denote phosphorylation and the catalytic cysteine; the current coloring scheme is confusing.*

We have modified the colours in this figure.

Reviewer 2

The model provides a hypothesis for the role of PINK1 phosphorylation of the Ubl domain and the role of phosphorylated Ub in achieving the fully active E2~Ub bound state of Parkin. The detailed residue-level examination of interactions via NMR methods enables a stepwise examination of the interfaces between the relative domains and rearrangements required to support these interactions. Correlation of the findings with functional assays and comparison to other data for mutations is strong. It is a well-constructed model built on clever deconstruction of the interactions. While the study encompasses some divide-and-conquer strategies, the authors are careful to use the most complete components of the systems as is feasible and which is relevant to the interactions under study.

The conclusions and the model are well supported by the data and provide new insights in to the activation process of Parkin. While there is more to be elucidated, this study lays out the outline of the process and will support further research by both these groups and others.

We thank the reviewer for their supportive comments and appreciation for our work.

Comments:

Page 7: Middle paragraph: suggest that the sentence should read "...RORBR shift in the presence of E2-Ub and ..."

This sentence was modified – currently near top of p7

Page 7, bottom paragraph: The input data for the HADDOCK calculation should be given some scale in the text, which is in addition to the methods. The number of restraints or contacts observed and used in the calculation should be provided. This would be a summary of the total in the text and in the respective categories in the Methods section (page 18).

We have revised this and included the number of active and passive restraints we used according to the HADDOCK protocol. On p9 we describe the starting co-ordinates we used (RORBR (5N2W), UbcH7 (PDB 4Q5E) and Ub (PDB 1UBQ)), selection of residues used to conduct the docking, number of restraints and docking approach.

Page 8, Figure 1D and Methods: The experiment reported in Figure 1D is insufficiently described. Is this a protection assay for binding of UbcH7-Ub to RING1 and IBR?

This was a peptide array that another reviewer thought was confirmatory. It has been removed from the manuscript.

Page 9 bottom paragraph: The observation of residue W403 as an anchor for the REP element should be confirmed by a significant effect on the indole sidechain resonance of residue W403. Is this observed? If not, why? Also, residue V465 is explicitly mentioned, but it does not appear to be visible in Panel B.

Unfortunately we did not have the indole sidechain resonance assigned. However, the backbone amide for W403 has shifted(broadened) beyond recognition upon UbcH7-Ub addition. We did not label V465 in Panel B because this signal is in a congested section of the NMR spectrum and a shift/broadening could not be unambiguously identified.

Page 9-10: Figure 2. The peaks indicated in Panel A correspond to residues that should be visible (at least some of them) in Panel B. The figure should be adjusted to highlight these residues in Panel B.

This figure is now Fig. 3. We have modified the colouring on Fig. 3B to highlight W403. In this figure many residues that shift (disappear) upon E2-Ub addition are shown in the spectra in Fig. 3A including S255, E399, D464, W403, S145, R234 and Q400.

Page 10: The postulation for losing signals in the RORBR:pUB UbcH7-Ub complex is that a remodeling occurs. Is this due to chemical exchange effects that result from indirect or allosteric effects? Is this seen at all molar ratios of the complex, or is it possible to observe the shift at a low molar ratio followed by disappearance at a higher molar ratio? Could the exchange effect also be confirmed by either a temperature or field dependence effect?

We conducted these experiments at different UbcH7-Ub concentrations. Generally signals that disappeared exhibited slow exchange effects, especially for the E2 binding region. This is consistent with the dissociation constant for E2-Ub with parkin ($< 1 \mu\text{M}$). We did not go to higher molar ratios due to difficulties with the samples. The reviewer is likely correct that that the remodeling is an allosteric effect although we did not use this term in the manuscript.

Page 11 bottom paragraph: Figure 4A/B the ratio of components is stated as equimolar for panel A but not stated for panel B. This should be specified to support and enable interpretation of the 'fast-exchange' condition stated in the text on page 11-12.

Please note this figure is now Fig 4C,D and the order of the panels has been reversed as requested by another reviewer. In the previous panel B (now Fig. 4C) there is no ratio of components. This experiment aims to detect a potential *intramolecular* interaction of the pUbl with the remainder of the parkin protein. This panel shows the isolated pUbl spectrum only (black contours) and the full-length phosphorylated parkin + pUb (green contours). The small changes in chemical shift we see are consistent with a weak intramolecular interaction, although it is not possible to determine what percentage would be bound here. However, the pParkin:pUb spectrum was collected using a 30 millisecond T2 filter such that a large complex (ie. parkin) would be nearly invisible in the spectrum. This approach was published last year by our group (PNAS, 2017). However, in this spectrum we see all the resonances for the pUbl domain (in the context of full-length parkin) indicating it can not be tightly bound to the remainder of parkin (or it would be invisible).

Page 12 and Figure 5B: The color coding in Figure 5B is not described. Presumably, it corresponds to the peptides with large HDX factors that are described in the text. Some color code needs to be provided in either the text or the figure legend.

This has been corrected in the figure legend. It is now Fig. 1.

Page 13: ...is blocked; however, static.... Use semicolon

Corrected

Page 13: The pdb codes for some of the structures should be listed in the figure legends or the text somewhere. The first paragraph of the Discussion refers to them, and there are multiple structures used to make the figures, so citation of the PDBID should exist somewhere in the manuscript. This is done in part in the methods, but the entire manuscript should be reviewed to for consistency on this point.

We have added PDB codes where appropriate throughout the text and in the figure legends.

Page 16-18: There are numerous reagents for which a source or description is not available. For example, UbVS, Ub800, etc.

This has been added in the Methods section.

Page 26: the legend to Figure 4 does not indicate the changes in the rest of the spectrum. Were these the only residues that shifted or were there some apparently non-specific effects? In panel B, the experiment is a mixture of ¹⁵N-labelled pParkin and pUb, but the molar ratio is not defined. This ratio should be included. Do the shifted peaks completely disappear at a higher ratio and/or are more peaks observed at a smaller ratio?

The experiment in panel B was done with equimolar amounts of pUb with pParkin. This complex was purified using a size exclusion column. The resonances indicated comprised nearly all signals where we saw any measurable shift. We have identified D62, Q63, Q64 and pSer65 in the spectrum. The other signals present (green contours) represent signals from flexible linker regions in parkin that are not part of the pUbl domain. We do not have these assigned and hence these are not labelled.

Page 27: Legend to Figure 5 B. The various colored segments in the structure are not described, nor are they described in the text. Since these colors are intended to reveal aspects of the interaction, then they should be described in either the main text or in the legend.

This information has been better described in the text and figure legend. This figure is now Fig. 1.

Supp. Fig 5: the domain names RING0, RING1, etc. are rather weak and would benefit from a bold

font.

This figure has been modified. It is now Fig. EV1.

Supp. Fig 7: add an x-axis label of (min)

Labels have been added. These are now Appendix Figs. S1, S6

Reviewer 3

Overall, this study provides some novel insights into Parkin mechanism, however, there are several important issues that need to be addressed in order to clarify and strengthen the author's major conclusions and elevate the manuscript to a state more suitable for publication in EMBO. Lastly, Gladkova et al., recently published a manuscript describing the structure of phosphorylated parkin that has several overlapping elements with this study. The data and conclusions from the Gladkova et al. study should be considered and discussed.

We were pleased to see your positive comments. The manuscript has been revised to consider the structural work from the recently published Gladkova and Sauvé papers.

Comments:

- *The recent Gladkova et al. paper describing the structure of phosphorylated parkin should be referenced. How the HADDOCK model and other data presented in this paper compare to PDB:6GLC and the HDX data presented in the Gladkova et al. study should be discussed.*

Both the Gladkova and Sauvé papers were released just after we submitted our manuscript. As they both provide important structural and HDX information we have highlighted this in the Discussion section of the revised paper.

- *Where were the backbone resonance assignments of the R0RBR Parkin fragment obtained? Surely these assignments were not trivial and this data underlies many of the CSP experiments, including the underlying data used to generate the HADDOCK model. However, there is no detail of how these assignments were performed/validated in the Methods, there is no citation in which these details were provided, nor is there an R0RBR entry in the Biological Magnetic Resonance Data Bank. Will the assignments and CSPs be deposited in the BMRB?*

The assignments for R0RBR were completed using standard TROSY-based triple resonance experiments as described in a previous publication (EMBO J, 2015). Consistent with many other groups (ie. parkin Ubl-endophilin SH3) that have simply used CSPs or relaxation methods to identify binding regions or conduct HADDOCK modelling we have not deposited our assignments nor the models. In the current work we have most assignments for the backbone (HN, N, CA) region of R0RBR so would be of little use to most groups but very few sidechain assignments. It is more typical that complete NMR assignments are deposited to the BMRB when one uses them to determine a three-dimensional structure of a protein, which is also deposited to the PDB. As a result, we have deposited the complete assignments for the pUbl (30197), RING2(Rcat)(18642), IBR (15074) and tandem IBR-RING2 (18990) domains to the BMRB as these were used to determine NMR structures of these domains. Of course we would be happy to make our assignments and HADDOCK models available to anyone who might find them useful. We have added a note in the manuscript regarding this.

- *Figure 3A,C. The author's model for why UbcH7-Ub binding decreases reactivity of parkin's catalytic cysteine in the context of full length pParkin/pUb/E2-Ub is hard to understand. Wouldn't this data suggest that pUbl interactions with RING0 attenuate parkin activity in the presence of UbcH7-Ub rather than 'act synergistically to drive parkin activity' as the authors state? Further, the total levels of UbV modification occurring in full length pParkin/pUb/UbcH7-Ub and R0RBR/pUb/pUbl/UbcH7-Ub are quite similar. If pUbl and UbcH7-Ub binding were acting synergistically to drive parkin activity wouldn't the total levels of UbV modification be expected to be greater in the context of full length pParkin? Might a kinetic analysis be more informative?*

This figure has been moved to Fig. 5 behind the new HDX experiments with UbcH7-Ub. In the text we hypothesized that the decrease in reactivity in full-length pParkin:pUb:E2-Ub might indicate the RING2(Rcat) has been repositioned (as our HDX experiments suggest) to a position alongside the E2-Ub similar to that observed for HOIP/UbcH5b-Ub. In this state, even if occupied transiently, the catalytic cysteine would be more protected and unable to recruit a Ub-VS molecule because it is near the donor Ub in the UbcH7-Ub conjugate. We also note that this experiment was not done in the Gladkova work. We have used the term synergy to reflect that the E2-Ub and pUbl work together to alter either pUbl binding (shown by NMR) and Ub-VS reactivity. For example, in the Ub-VS reactivity, the co-addition of pUbl and E2-Ub to R0RBR:pUb or the addition of E2-Ub to pParkin:pUb gives a different result than pUbl added alone or pParkin alone. It may not be as expected but clearly the two ligands affect each other. In our Fig 5 the level of Ub-VS modification is always greater in full-length parkin. However, we feel that the R0RBR case is better to assess the pUbl interaction because it is intermolecular and shows increased activity in the presence of both pUbl and E2-Ub. We have done our assays using several timepoints which is more rigorous than typically done for this assay. While not true kinetics our data (Fig. 5) does show that our approach and data is consistent regardless of which time point used.

• Were the experiments in Figure 3 performed in triplicate? Were C431A negative controls performed to ensure the crosslinks are specific to the catalytic cysteine, particularly since there are several Ub binding sites? Were these experiments performed with the addition of free UbcH7? This latter experiment could provide some interesting details into the mechanism of the author's observations. Are the same results obtained in the presence of UbcH5-Ub?

These experiments are now presented in Fig. 5. They were done in duplicate and error bars are noted in panels B and D. As suggested we have now added negative controls in Appendix Fig. S7. Rather than using C431A we used C431F which is a Parkinson's ARJP substitution. We also conducted experiments using *Drosophila melanogaster* RING2(Rcat) alone where the equivalent C449 and C449F were tested. (Appendix Fig. S7). We did not conduct the same experiments with free UbcH7 as it has a weaker affinity for parkin. The suggestion of UbcH5b-Ub is interesting. However we felt this was outside the scope of this project given that UbcH5b has much lower ubiquitination activity with parkin.

• Supplemental Figure 1 displays a line that indicates the average CSP +/- 1 standard deviation, however, many of the residues discussed in the manuscript fall well below this threshold. What are the criteria by which the authors determined 'significance'?

This figure has been modified and is now Fig. S4. We have included more details regarding selection of residues from NMR data in the Results (p 9) and Methods (p25) sections and in the appropriate figures. Residues that “disappeared” due to large chemical shift changes (grey lines) and those that exhibited CSPs > one standard deviation above the average were selected (“active”) and filtered for accessible surface area according to the HADDOCK protocol. Other residues were selected (“passive”) because they were accessible and neighbored “active” residues, again according to the protocol. We did not have CSP information for some of these residues so these may be the ones the reviewer is referring to.

• Along these lines, the region between Parkin residues 230 and 240 harbors one of the largest clusterings of residues that experience CSPs above 1 standard deviation upon UbcH7-Ub binding. What accounts for the large chemical shifts in this region? Do these residues contact UbcH7 or Ub in the HADDOCK model? Were they included as 'interacting residues' during the HADDOCK calculations?

We have modified this figure (now Fig EV2) to annotate the CSPs in R0RBR and categorize them into those near the E2 or Ub binding sites or re-modelled (RM) sites. During our initial docking of UbcH7 with R0RBR we used all residues based on those shown in the figure. The resulting models consistently showed UbcH7 in the “bound” position near the canonical loops and helix in RING1 even though restraints for the re-modelled residues were included. This observation is what lead us to hypothesize that these additional residues were a result of re-modelling rather than binding which of course from the CSP information alone we can not determine. The residues indicated (230-237, no CSP information for 238-240) lie in the linker connecting the RING0 and RING1 domains near W403. They do not contact UbcH7.

• *The most novel aspect of the HADDOCK model is the parkin binding site for Ub from the Ubch7~Ub thioester mimetic. While the author's tested ARJP and non-ARJP mutations on the parkin side and found a loss of parkin activity, no Ub mutants at the predicted parkin/Ub interface were presented to validate the position of Ub in the model through either loss of function or gain of function in the context of the above parkin mutants. In the absence of biochemical data in support of the proposed Ub binding site, is it possible to perform the CSP experiments with free Ubch7 to determine whether the CSPs resulting from contacts between Ub and the putative Ub binding site on parkin are present? In addition to providing evidence in support of the author's HADDOCK model, this experiment would potentially illuminate the role of E2 binding versus Ub binding in triggering remodeling of the RING0/RING2 interface*

Thank you for this excellent suggestion. We have now completed new Ub loading experiments using a substituted parkin protein (C431S+H433A) that allows Ub transfer from the E2~Ub conjugate to the catalytic site in the RING2 (new Fig 2D). We chose substitutions in the Ub molecule (K6A, L8A, K11A) based on contacts from our structure. These substitutions impaired Ub loading to parkin and Miro and support the position of the Ub molecule. We have also added a section in the text (p9) to indicate that there was minor variations in the orientation of the Ub in Ubch7-Ub possibly due to dynamics and in agreement with its weak native affinity. As a result we show a new figure (Fig EV3) that shows the variation in Ub position (in the Ubch7-Ub conjugate). We have added CSP experiments with Ubch7 alone to R0RBR:pUb (Fig EV2). These were not as impressive as Ubch7-Ub binding likely due to weaker affinity (EMBO, 2015; NSMB, 2017) but generally agreed with the mapped site on the RING1 domain. Based on these observations we proposed that it is the Ub in the conjugate that is key to the remodelling.

• *Martino et. al recently demonstrated that Ubch7-Ub engages the RBR HHARI/ARIH1 differently from Ubch5-Ub. Have the author's performed the CSP experiments with Ubch5-Ub?*

This is an excellent suggestion. In our hands and in published work Ubch5 provides much lower ubiquitination activity with parkin than Ubch7. It would still be interesting to know why on a structural level. We hope to pursue these experiments separately in the future.

• *Page 14, third sentence from the bottom- issue with logic? The author's posit that rapid association/dissociation of pUbl with R0RBR:pUb is to ensure parkin activity only in the presence of the loaded E2~Ub, but parkin can only be active upon charging with Ub after E2~Ub has already bound.*

The Discussion section including the comments on p14 has been completely re-written to clarify this and several other statements. In essence we feel that the E2~Ub is doing more than simply delivering the cargo. We feel it helps with pUbl recruitment to the RING0 domain and it may be involved in RING2(Rcat) positioning. Our logic is consistent with the recent Gladkova and Sauvé papers that shows the pUbl bound. Further, our new HDX experiments show that the RING2(Rcat) exchange is influenced by the Ubch7-Ub.

• *The last sentence of page 8 leading into page 9 is quite difficult to follow and the accompanying figure does not clearly show what is being stated in the text. Also, what putative interactions lock the flexible Ub C-terminus in an orientation in which G76 carboxyl group is 'exposed' towards the tether side of parkin? There do not appear to be any parkin residues in proximity.*

This section is now on p10 (bottom). We have modified the text and removed the accompanying figure. While the overall arrangement of the E2-Ub could be determined we did not dwell on specific interactions near the Ub C-terminus because this is a model rather than an x-ray structure. Due to the HADDOCK modelling protocol the orientations of side chains in parkin Ubch7 and Ub are largely similar to those in the starting structure so we felt that detailing interactions at the requested level would be over-stepping our boundaries.

• *The abstract is misleading. 'We present the structure of an incoming E2~Ub conjugate with phospho-ubiquitin bound parkin R0RBR' should be clarified. The structure that is presented is a HADDOCK model rather than an experimentally derived structure. The title of Figure 1 or the*

sentence on page 7 stating "We used chemical shift perturbation results to determine a model of a Ub_{CH7}-Ub bound to R0RBR:pUb using HADDOCK" is more accurate.

We have clarified wording in the Abstract to reflect this.

• Typo page 8, line 11? Is V293 correct?

This has been corrected in the text.

2nd Editorial Decision

24th October 2018

Thank you again for submitting your revised manuscript for our consideration. It has now been seen once more by two of the original reviewers, and I am happy to inform you that they are both satisfied with the experimental and presentational improvements of the paper. At this stage, both reviewers only have a few specific comments that I would like to ask you to incorporate in a final round of minor revision, which should also take care of a number of remaining editorial issues.

REFEREE REPORTS

Referee #1:

The revised manuscript is greatly improved. Reorganization of the order in which data are presented, plus additional data not included in the original manuscript (HDX data in Fig. 4 and NOE data in EV1) help to clearly lay out the new insights and contributions from these studies. The authors now do a good job of comparing their results to those recently published and highlight both the similarities and differences. This greatly helps the reader to appreciate and understand what is otherwise a complicated system and story. Although the manuscript could be published in its present form, there remain a few instances where some clarification would be helpful. These are minor and will not change the message or story.

MINOR Corrections

1. P. 10. The authors cite Dove et al., 2017 and Yuan et al., 2017 in support of an open E2~ Ub binding to HHARI. That observation was first established by Dove et al., 2016 using NMR. The later two papers confirmed this. As an NMR spectroscopist himself, I'm sure that the corresponding author appreciates the primacy of the NMR result.
2. p. 12. Some care needs to be taken when describing/discussing the orientation of structural elements within the HOIP/UbcH5 complex (Lechtenberg et al., 2016) as some of the components that contact each other are intermolecular due to a "domain-swap" in the crystal. While such contacts are likely relevant, it is important not to let that "experimental detail" get forgotten so that subsequent readers (who may not go back to the original paper) fail to understand that point. I know it's a bit tricky in the writing, but I feel strongly that the point needs to be made.
3. P. 12, 2nd para. The species being trapped in the experiment using C431S/H433A is an oxyester. Although "ester" as written is correct, specifying the oxyester is helpful to non-expert readers.
4. p. 13. Two spelling issues: "lessor" and "complimentary"
5. p. 14, last para. The second sentence begins "On the surface..." Because the first sentence is about exchange, I thought that the "surface" in the second sentence related to the protein surface and had to re-read the sentence several times to understand it. This may just be my quirky reading, but the authors may wish to consider a different choice of words here.
6. p. 20 middle paragraph "...an addition segment" should be "...an additional segment"?

Referee #3:

In their revised manuscript, Condos et al, have suitably addressed the majority of this reviewer's key questions and concerns. In strengthening their major conclusions through additional experiments and analysis, this work now represents a significant advance in our understanding of the structural basis for Parkin activation. As such, I believe the manuscript is well-suited for publication in EMBO. My last comment is that the authors should include the pdb coordinates of the lowest energy R0RBR:pUb/UbcH7-Ub HADDOCK model in the supplementary information. The model is key to many aspects of the study and free access to coordinates would allow readers (particularly for non-structuralists) to interpret the model and follow the text in ways that might be difficult solely based on the panels presented in the manuscript.

2nd Revision - authors' response

25th October 2018

Point-by-point response

R3 asks that we include the pdb coordinates for our model as Supplemental. I could submit to the PDB and add the PDB accession during proofing.

I've changed the Data Availability as suggested.
All revisions completed. Abstract shortened to 175 words.
All files uploaded.

Accepted

26th October 2018

Thank you for submitting your final revised manuscript for our consideration. I am pleased to inform you that we have now accepted it for publication in The EMBO Journal.

Corresponding Author Name: Gary S. Shaw

Journal Submitted to: The EMBO Journal

Manuscript Number: EMBOJ-2018-100014